



# Modeling interactions between tides, storm surges, and river discharges in the Kapuas River delta

Joko Sampurno[1,2], Valentin Vallaeys[1], Randy Ardianto[3], Emmanuel Hanert[1,4]

[1]Earth and Life Institute (ELI), Université Catholique de Louvain (UCLouvain), Louvain-la-Neuve, 1348, Belgium
[2]Department of Physics, Fakultas MIPA, Universitas Tanjungpura, Pontianak, 78124, Indonesia
[3]Pontianak Maritime Meteorological Station, Pontianak, 78111, Indonesia
[4]Institute of Mechanics, Materials and Civil Engineering (IMMC), Université Catholique de Louvain (UCLouvain), Louvain-la-Neuve, 1348, Belgium

*Correspondence to*: Joko Sampurno (joko.sampurno@uclouvain.be, jokosampurno@physics.untan.ac.id)

**Abstract.** The Kapuas River delta is a unique estuary system on the west coast of Borneo Island, Indonesia. Its hydrodynamics is driven by an interplay between storm surges, tides, and rivers discharge. These interactions are likely to be exacerbated by global warming, leading to more frequent compound flooding in the area. The mechanisms driving compound flooding events in the Kapuas River Delta remain, however, poorly known. Here we attempt to fill this gap by assessing the interactions between river discharges, tides, and storm surges and how they can drive a compound inundation over the riverbanks,
particularly within Pontianak, the main city along the Kapuas River. We simulated these interactions using the multi-scale hydrodynamic model SLIM. Our model correctly reproduces the Kapuas River's hydrodynamics and its interactions with tides and storm surge from the Karimata Strait. We considered several extreme scenario test cases to evaluate the impact of tide-storm-discharge interactions on the maximum water level profile from the river mouth to the upstream part of the river. Based on the maximum water level profiles, we could divide the main branch of the Kapuas River's stream into three zones, i.e., the
tidally-dominated region (from the river mouth to about 4 km upstream), the mixed-energy region (from about 4 km to about 30 km upstream) and the river-dominated region (beyond 30 km upstream). Thus, the local water management can define proper mitigation for handling compound flooding hazards along the riverbanks by using this zoning category. The model also successfully reproduced a compound inundation event in Pontianak, which occurred on 29 December 2018. For this event, the wind-generated surge appeared to be the dominant trigger.

## 1. Introduction

Global warming leads to more frequent tropical storms, rising sea levels, and more intense rainfalls, which all concur in increasing the occurrence of compound flooding events in a coastal area and its surrounding environment (Bevacqua et al., 2020). Such disasters are impacting the lives of more and more people worldwide (Moftakhari et al., 2015). Compound flooding occurs when dry low-lying land (over a coastal area) is flooded by water from the ocean and the river. Compound
flooding is driven by the interaction between a coastal inundation and a riverine inundation. The former happens from the





seaward direction due to high (spring) tides and storm surges, while the latter occurs from the landward due to high discharges from upstream rivers. Coastal communities, which have been growing in population over the past decades, have become increasingly vulnerable to those events. Cities located along an estuary are at the crossroad between the ocean and the river catchment, hence particularly vulnerable (Herdman et al., 2018; Vitousek et al., 2017; Zhang and Liu, 2017).

One of the most significant drivers that can trigger compound flooding over a coastal area are storm surges (Herdman et al., 2018; Zijl et al., 2013). A storm surge is defined as the difference between the observed water level and the expected water level that results from tidal dynamics in a coastal area. A storm surge effect on usual tidal dynamics is the altered timing of high and low water through non-linear processes (Zijl et al., 2013). A storm surge is composed of low- and high-frequency components (Spicer et al., 2019). The former modifies the non-tidal water level, and the latter represents a tide-surge interaction

during the event. A storm surge is generally quantified by the skew surge (Giloy et al., 2019), which is a tidal cycle average measurement—the difference between the observed height and the expected height. It reflects the level of surge generated over a tidal cycle. To mitigate compound flooding hazards in a coastal area, storm surge is a critical component that should always be taken into account.

Several factors must be considered regarding the assessment of compound flooding risks in a river delta. The first factor is the

coincidence of the sources (Herdman et al., 2018), which means there is a possibility for the delta to receive an extreme flow from the upstream, while, at the same time, there could be an intense surge occurring in the tide or excessive rainfall over the coastal area. The second factor is the dependency and interdependency of the sources, indicating whether the interaction between these sources (extreme flows, tides, and excessive rainfall) could significantly impact the inundation processes (Bilskie and Hagen, 2018; Herdman et al., 2018; Santiago-Collazo et al., 2019). Other important factors include the vegetation

properties along the riverbanks that resist the flow of water; the vegetative properties over the estuary that reduce the momentum transfer of wind; the landscape characteristics of the coastal area; and how they interacted with each other (Twilley et al., 2016).

Flooding events in a delta area can be simulated and assessed using hydrodynamic models (Deb and Ferreira, 2017; Olbert et al., 2017; Patel et al., 2017). The most useful models are those that can seamlessly simulate the processes occurring along the

land-sea continuum, which corresponds to the area encompassing the coast, the estuary, and the river channels. Such models allow for a study of past flooding events and an assessment of flood mitigation strategies (Vu and Ranzi, 2017). However, realistic input variables and forcing are needed to create an accurate physics-based hydrodynamic model. Since the processes that drive compound flooding are three-dimensional, a 3D model is the most appropriate tool to evaluate the event. However, the use of a 3D model generally requires high computational costs. As an alternative, two-dimensional models are a good

compromise between physical realism and computational costs, particularly for shallow and well-mixed domains (Huybrechts et al., 2010; Néelz, 2009). By reducing the physical complexity of the model, it is possible to increase the horizontal resolution; and hence, the model will better represent the system topography.

As an archipelago country with about 100,000 km of coastlines, Indonesia is faced with significant coastal flooding risks (World Meteorological Organization, 2019). Indonesian coastal flood hazards are classified as high (Fraser et al., 2016). It





means that the potentially damaging waves are expected to flood the coasts at least once every ten years. Based on this risk
and concerns about the impact of climate change in the future, a hazard assessment study in Indonesian low-lying coastal areas
(such as deltas) is become critical for Indonesian water management authorities. One area in Indonesia that is vulnerable to
coastal or even compound flooding is the Kapuas River delta. In this delta, flooding events happen more than once a year
(Wells et al., 2016). Unfortunately, there is no previous study addressing the process that underlies the flood events in there.

The previous study (Hidayat et al., 2014) successfully evaluated the inundation frequency, but only for the upstream area and
did not evaluate the underlying process. Therefore, this study attempts to fill in the gap and provide the first compound flood
assessment in the area.

This paper investigates the interaction between tides, storm surges, and river discharges in the Kapuas river estuary and its
surrounding area. We use a 2D hydrodynamic model to simulate the interaction between these driving forces and their effects

on the compound flooding in the Kapuas River delta, particularly in Pontianak. Then, we create a detailed flood assessment,
determine the area's extent, and calculate the inundation depth. As a case study, we implemented the model explicitly to a
compound flood event on 29 December 2018.

## 2. Material and Method

### 2.1 Area of interest

The Kapuas River flows from the center of the Borneo island (Indonesia) toward the Karimata Strait on the west coast (Fig.
1). The river is one of the longest island's rivers in the world, with a length of about 1,143 kilometers (Goltenboth et al., 2006).
The Kapuas river basin is located on the equator with high air temperature and humidity throughout the year. The river basin
spreads over about $8.28 \times 10^4 \, km^2$, with about 66.7% of it consisting of forests (Wahyu et al., 2010). The topography of the
river comprises hills over its upstream and plain over its downstream. The river flow ends at the Karimata Strait, creating a

five-arm delta in its estuary (MacKinnon et al., 1996).

The largest distributary of the Kapuas River is the Kapuas Kecil River. The river starts from a Kapuas River branch at the
Rasau Jaya district and ends at the estuary area in the Jungkat district. In the middle of its streamflow (about 20km from the
river mouth), the river flow creates a junction with the end stream of the Landak River. The junction is located in the city of
Pontianak, the capital of West Borneo Province, Indonesia, placing the urban area at the highest risk of flooding among other

areas along the Kapuas riverbanks.

Pontianak is the most populated urban area on the west coast of Borneo island, with a population of about $6 \times 10^5$ people that
keeps increasing. The city is located on low-lying land and has 61 canals spread across the area. Most of these canals flow in
the Kapuas Kecil River (Pemerintah Kota Pontianak, 2019). As a consequence of its geographical situation, the city often
experiences inundations.





## 2.2 Hydrodynamic model


The interplay between the river discharges, the tides, and wind surges from the sea and their effect on the inundation processes in Pontianak is investigated by using the Second-generation Louvain-la-Neuve, Ice-ocean Model (SLIM, https://www.slim-ocean.be/). The model equations are discretized with the discontinuous Galerkin finite element method. The model uses an unstructured mesh, whose resolution can vary in space. The model has successfully been applied to several areas, such as the

Great Barrier Reef (Lambrechts et al., 2010), the Mahakam Delta (Pham Van et al., 2016), the Scheldt Estuary (Gourgue et al., 2013), and the Columbia River (Vallaeys et al., 2018).

Here, we use the wetting-drying barotropic version of SLIM that solves the conservative form of the Shallow Water Equations (SWE):

$$\frac{\partial H}{\partial t} + \nabla \cdot \boldsymbol{U} = 0 \tag{1}$$

$$\frac{\partial \boldsymbol{U}}{\partial t} + \nabla \cdot \left(\frac{\boldsymbol{U}\boldsymbol{U}}{H}\right) + f\boldsymbol{e_z} \times \boldsymbol{U} - \nabla \cdot (\upsilon \nabla \boldsymbol{U}) = \alpha g H \nabla (H - h) - \frac{Cd}{H^2}|\boldsymbol{U}|\boldsymbol{U} + \frac{1}{\rho}\tau_{wind} - \frac{H}{\rho}\nabla p_{atm} \tag{2}$$

where $\boldsymbol{U} = H\overline{\boldsymbol{u}}$ is the horizontal transport, $H$ is the water column height, $h$ is the bathymetry, $t$ is the time, and $\overline{\boldsymbol{u}} = (u, v)$ is the depth-averaged horizontal velocity. $\alpha$ is a constant that is set to zero over dry elements and one over wet elements (Le et al.,

2020), $\nabla$ is the horizontal gradient operator, $g = 9.81$ m/s$^2$ is the gravitational acceleration, $Cd$ is the bulk drag coefficient, $f$ is the Coriolis parameter, $\boldsymbol{e_z}$ is the vertical unit vector pointing upward, $\upsilon$ is the horizontal eddy viscosity, $\tau_{wind}$ is the wind stress, and $\nabla p_{atm}$ is the atmospheric pressure gradient.

The model equations are solved using a wetting-drying algorithm with an implicit time-stepping scheme. With this algorithm, a mesh element can be defined as wet or dry. It requires a procedure that guarantees that the water column height ($H$) is always

positive at the end of the time step to solve the equations. To achieve that, we set first a threshold $h*$ that indicates the water height below which an element is assumed to be dry (here $h* = 0.5$ m). Then, we define a parameter representing the total water column height on an element as a combination of maximum water level and minimum bathymetry: $s = \max(\eta) + \min(h)$. When $s$ is smaller than $h*$, gravity will be canceled on the element so that the artificial gradient of surface elevation would not remove water from an already-dry element ($\alpha = 0$). On the contrary, if $s$ is greater than $h*$, then the gravitational force will be

preserved as usual ($\alpha = 1$). The transition between $\alpha = 0$ and $\alpha = 1$ occurs smoothly. More details about this procedure can be found in Le et al. (2020).

## 2.3 Model setup

Before creating the model, we first define a domain covering both the ocean and the Kapuas River Delta. Then, we mesh the domain, set the bathymetry and the bulk drag coefficient, define the boundary conditions (surface elevation and velocity), and

finally impose some forcings. After the model is created and run successfully, we validate the results using observational data.





Here, we generate a multi-scale mesh of the entire domain by using the mesh-generation algorithm of Remacle and Lambrechts (2018). The mesh consists of 206,359 triangular elements. Fine mesh elements are used to accurately represent inundation events over the land area of Pontianak and along the Kapuas riverbanks, while the coarser mesh elements are used far away from the delta. The mesh resolution reaches 50 m along the Kapuas riverbanks and over the city of Pontianak. It decreases to
10 km in the middle of the Karimata Strait (Fig. 2).

The bathymetry is created from a combination of three different data sources. The first is the river and estuary bathymetry obtained from the Indonesian Navy (Kästner et al., 2019) with a $100 \times 100$ m grid resolution. The second is the Karimata Strait bathymetry, obtained from BATNAS (Badan Informasi Geospasial, 2018) with a $180 \times 180$ m grid resolution. The last is the digital elevation model (DEM) from SRTM (NASA JPL, 2013) with a $30 \times 30$ m grid resolution. The Karimata Strait
bathymetry shows that the strait is shallow (less than 100 m) and relatively flat (Fig. 3). On the other hand, the bathymetry of the river is more heterogeneous. The river is shallow in the estuary but deeper in the middle stream. The depth ranges from 1 m (in the estuary area) to 40 m in the middle stream area (Kästner et al., 2017). The Kapuas Kecil River, which flows through Pontianak, has a depth that decreases from 15 m in the middle stream to 1 m in the river mouth.

The wind velocity and the atmospheric pressure data are the ERA5 reanalyzes dataset obtained from the European Canter for
Medium-Range Weather Forecast (ECMWF) (Hersbach et al., 2020). They have a spatial resolution of $0.125° \times 0.125°$, while the temporal resolution is three hours. Unfortunately, compared with the marine weather station's observational data, there is a difference in its peak magnitude. The observed wind velocity is more significant than the wind velocity from ERA5 during a wind surge (Fig. 4). It may happen because the time resolution of the ERA5 wind data is coarser (3 hours) compared to the observational data resolution (hourly) so that the ERA5 data does not capture short-period wind surge events. Therefore, in
the case study, we adjust the magnitude of the wind input data (ERA5) during the wind surge event. We multiplied the wind magnitude with a ratio between both peaks (the observed and ERA5 data) at the observation point close to the river mouth.

Next, we set the bulk drag coefficient to $2.5 \times 10^{-3}$ over the ocean part of the domain and to $6.9 \times 10^{-3}$ over the river part. The former corresponds to a sandy sea bed, while the latter corresponds to a muddy river bed (Li et al., 2004). Over the dry area, the drag coefficient is determined by two types of land cover (urban and non-urban area), as defined in the Copernicus
Global Land Cover map (Buchhorn et al., 2020). For the urban area, the bottom drag coefficient is 2.05 (Hashimoto and Park, 2008). Over the non-urban area, it is set to 2.0, which corresponds to dense vegetation (Li and Busari, 2019).

Upstream of the rivers, we imposed the discharge of the Kapuas River and the Landak River. Since there are no observational data for both rivers, we set constant discharges for the simulations in some scenarios. However, to reconstruct the observed inundation (discussed next in the case study), we imposed the discharges obtained from the Global Flood Monitoring System
(GFMS). GFMS estimated the global discharge based on TRMM Multi-satellite Precipitation Analysis and Global Precipitation Measurement (Wu et al., 2014). The Kapuas River discharge is much larger than the Landak River discharge (Fig. 5).

To evaluate the model performance, we performed the tidal analysis on the simulated surface elevation and compared it to the observed water level at the tide gauges. The analysis was conducted using the Unified Tidal Analysis (UTide) Python package





(Codiga, 2011). The tidal analysis assumes that a tidal signal is a linear combination of multiple sinusoidal signals associated
with its astronomical forcing. Each of these signal components has a different phase, amplitude, and temporal frequency. The
analysis aims to obtain tidal harmonic constituents. Since we only ran the model for a limited time duration (in order to
minimize the computational cost), it is impossible to extract all tidal constituents from the model's signal output. As a rule of
thumb to select the appropriate tidal constituent, we used the Rayleigh criterion (Godin, 1970). Based on this criterion, we

obtained six tidal constituents with satisfying frequencies, i.e., K1, O1, Q1, J1 (as the representation of the diurnal components)
and M2, S2 (as the representation of the semidiurnal components).

To perform tidal analysis, water level signals were extracted and analyzed at two points. The first point is located at the river
mouth, and the second point is located along the Kapuas Kecil River within the city of Pontianak (Fig. 1). The second point is
located about 20km from the river mouth. At the first point, the model's output signal is compared with the tidal signal from

the Indonesian Consortium of Oceanic and Atmospheric Prediction (ina-COAP) (Pusat Jaring Kontrol Geodesi dan
Geodinamika, 2014). Meanwhile, for the second point, the model's signal output is compared to the observational data
collected by the Pontianak Maritime Meteorological Station (PMMS).

To quantify the agreement between the simulated water level and observation, we calculated the Pearson correlation coefficient
(PCC), Nash-Sutcliffe Efficiency (NSE), and Root Mean Square Error (RMSE) between both time series along the river within

the city. The former coefficient measures the linear correlation between the model and the observation time series yields a
value between 0 (no correlation) and +1 (perfect correlation). The second coefficient measures how well the model's
performance represents the observed data. NSE = 1 represents a perfect model, NSE = 0 depicts a model with a predictive skill
the same as the mean of the observed data, and NSE < 0 informs that the mean of the observed data is a better predictor than
the model output. The latter coefficient indicates the average of the difference of the peaks between both time series.

**3. Results**

**3.1 Model validation**

The validation results show a good agreement between the model output and the ina-COAP's tidal signal at the Kapuas Kecil
river mouth, with a coefficient of determination $R^2 = 0.95$ and RMSE = 0.09m. At this point, the model produces a tidal signal
that has constituents attribute (phase and amplitude) similar to the ina-COAP's tidal signal. The diurnal components explain

about 95% of the total variances of the estuary's tidal dynamics, while the semidiurnal components only account for about 5%
(Fig. 6 and Appendix Table A1 for more detail). Next, the tidal analysis result at the observation point in Pontianak also shows
a good agreement between the model output and observation with a coefficient of determination $R^2 = 0.83$ and RMSE = 0.14m.
The model output signal also has similar constituents attributed to the PMMS observational water level signal (Fig. 7 and
Appendix Table A1). The analysis at this second point also confirms that the model successfully reproduces the observed

hydrodynamics.



### 3.2 The impact of river discharges on the river's maximum water level.

The impact of the Kapuas River upstream discharge on its downstream hydrodynamics is assessed by imposing different discharge values ranging from 3,000 m$^3$/s to 9,000 m$^3$/s (the highest discharge set based on the observed data in 2018, see Fig. 5). At the same time, the Landak River upstream discharge is set to 300 m$^3$/s for all scenarios. The maximum water level (MXWL) at different points along the Kapuas Kecil River stream is then calculated for these different discharge scenarios (Fig. 8). Based on these MXWL profiles, we thus know that the river discharge fully controls the maximum water level from the upstream river input to about 30 km before the river mouth. The MXWL profile linearly increases with the discharges. The water level fluctuation in this stream part is therefore river-dominated. Then, from this point to a point located at about 4 km from the river mouth, all MXWL profiles start to drop, indicating that there is a compounding effect of the discharge and the tide. The MXWL profile does not increase linearly due to the increase of discharge from the upstream. The different discharge scenarios have a different MXWL distribution pattern in this area. In the lowest discharge scenario, the MXWL profile decreases until about 6 km before the river mouth. Then, it starts to increase gradually and is stable at 4 km before the river mouth. On the other hand, in the highest scenario, the MXWL profile constantly decreased until a point 4 km before the river mouth, then remained steady. In addition, the closer the flow to the river mouth, the lower the gap between the MXWL profiles. This stream part, where both the river discharge and the tide control the MXWL, is called a mixed-energy region. Meanwhile, at about 4 km from the river mouth, the discharge variations have almost no influence the MXWL. In this area, the tides fully control the MXWL. Therefore, this part of the river can be characterized as tidally-dominated.

### 3.3 The impact of wind surges on the river's maximum water level

To evaluate the storm surge effect, we consider wind velocity scenarios. For each scenario, we multiplied the average wind velocity (about 4.9 m/s over the whole area) during the inundation event by a constant value ranging from one to five (W1 to W5 scenarios). Then, we evaluated their impact on the MXWL profiles. Based on the profiles (Fig. 9), we thus know that the storm surge strongly dominates the MXWL from the river mouth to about 4 km up the river. The increasing wind velocity directly increases the MXWL distribution profile. The stronger the wind velocity, the higher the MXWL is. Furthermore, from about 4 km to about 7 km of the river mouth, if the wind velocity is set to more than three times its average, the MXWL profile will drop along this area. The stronger the wind velocity, the sharper the decrease is. The decline of the MXWL profile could be due to the fact that the water levels within the river are higher than its riverbanks and then outflows to the floodplain. The outflow to floodplains leads to a reduction in the water levels in the main river.

### 3.4 A case study: analysis of the flood event on 29 December 2018

Based on the water level observation at the Pontianak Maritime Meteorological Station (see Fig. 1 for the location), on 29 December 2018, there was a significant increase in water levels from 05.20 UTC to 07.50 UTC. The maximum water level reached 2.8 meters and produced a significant inundation throughout the city of Pontianak. In order to investigate the main



drivers of the event, we simulated the hydrodynamical process along the land-sea continuum for the full month of December 2018.

The validation result shows that the PCC of both simulated and observed water levels in the middle of the city is 0.91, which
means that both time series are well synchronized. The NSE is 0.82, indicating that the model has a good performance. The RMSE of its peaks is 0.04 m. These results suggest that the model correctly reproduces the actual hydrodynamic processes. The inundation event on 29 December 2018 is well depicted by the model (Fig. 10, light grey box area). The observational data profile (black) shows that the water level dynamic is at the peak moment during the event, where its top seems lower than the previous peak period. The water level dynamic is about to go down when suddenly a strong force pushes it to go up again
for a short moment. After this additional forcing effect disappeared, the water level then decreased steeply and started following the tidal signal again.

A more qualitative validation of the model has also been performed by comparing the extent of the simulated inundation area with observations reported by the local media (Madrosid, 2018). The report mentioned some areas that were confirmed as inundated at the moment when the flooding occurred (represented by the red dots in Fig. 11). Fig. 11a shows that before the
inundation occurred, these areas were dry. Then, at 07.00 UTC, these areas became wet and confirmed inundated (Fig. 11b). At the time, the overtopped water flooded densely populated areas in the city, located about one block from the riverbanks. We further calculate the inundation extent and depth quantitatively. The former is calculated by estimating the total area of dry elements that become wet. The latter is calculated by summing up the maximum water level and minimum bathymetry over the inundated elements. As the result, we thus know that the flood event occurred during a short period. It started at 02.00
UTC, peaked at 06.00 UTC, and finished at 10.00 UTC. At the peak moment, the area inundated was about 9.78 km² (Fig. 12a), with a depth reaching 1.5 m. The extent of this inundation reached only areas close to the riverbanks and small low-land regions in the southwest of the city connected by a canal (Fig. 12b).

## 4. Discussion

Based on the discharge scenarios, we found that the city of Pontianak is located in the mixed-energy region. The maximum
water level of the river in the city is therefore controlled by the interaction between discharges from the upstream, storm surge, and tides from the ocean altogether. Consequently, to assess the inundation hazard in the city, the compound flooding scenario approach becomes more appropriate than only single-source flooding scenarios (only coastal, urban, or riverine flooding). Next, we also found that an extreme eastward wind velocity with a magnitude greater than 9 m/s can drive the seawater into the river channels based on the wind surge scenarios. As the storm surge continues to push the water upstream, the pile-up
water in the estuary will enter the river stream and propagate over several kilometers. If the water propagation is in cooccurrence with high river discharge, the water level can hence overtop riverbanks and lead to a compound inundation over the floodplain.





Furthermore, the wind velocity less than 9 m/s or more than 24 m/s (we simulated it but did not show the result here), it does not impact the MXWL profile in the mixed-energy region anymore. These thresholds become minimum and maximum wind
velocities that impact the MXWL profile inside the zone. In the thresholds, the change of the MXWL profiles between the wind velocity in 3$^{rd}$ scenario (wind speed = 14.7 m/s) and 4$^{th}$ scenario (wind speed = 19.6 m/s) is somehow more significant than the other MXWL profile changes. Unfortunately, we failed to define the zone border between river-dominated and mix-energy regions with this wind scenario. To obtain this border under the wind surge effect, we need to extend the river domain further upstream. We leave this to a future study.

On 29 December 2018, rainfall over the city of Pontianak was less than 7 mm, so that the effect of excessive rainfall could be ruled out in this event. Meanwhile, an intense wind velocity was observed over the coastal area for a few hours. The radar data shows Cumulonimbus convective clouds formed and moving eastward from the Karimata Strait towards the land (see Appendix Fig. B1). Cumulonimbus is a cloud type that could produce wind surges, tornadoes, and excessive rainfall (Cotton et al., 1992). These clouds reached and covered the Kapuas River's mouth on the date, from 04.30 UTC to 05.25 UTC.
Therefore, these clouds most likely triggered a storm over the coastal area with a wind speed ranging from 13 m/s to 21 m/s. The wind direction was oriented from the west to the east during the event. Hence, the total wind effect became the combination of the direct wind stress effect and the indirect wave run-up effect. Consequently, it drove the water column from the ocean to the west coast of Borneo island, where the Kapuas estuary is located. Then, the wind piled up the tidal waters inside the Kapuas river mouth. The piled-up water then propagated from the river mouth to the upstream and was coincidentally met with a high
river discharge. This phenomenon created an increasing water level and caused a short-duration overflow over the floodplain. Therefore, using this scenario, we suggest that the main trigger for the compound inundation, which occurred on 29 December 2018, was an interaction between a storm surge at the estuary area and a high discharge from the Kapuas River.

As with all modeling studies, there are some limitations related to the model that should be mentioned. Firstly, many different compound flooding schemes possibly occur in this area that has not yet been simulated. Therefore, the delineation of the stream
zones proposed for the Kapuas Kecil River needs further investigation in the future. Secondly, we have not yet imposed excessive rainfall scenarios in the model so that the model only depicts inundation processes as the effect of the river discharges, tides, and wind surge. In the actual case, a single excessive rainfall is enough to trigger urban flooding over the floodplain area. Next, the computational domain does not cover the whole dry land over the delta. It is only limited to Pontianak and the Kapuas Kecil riverbanks from the city to the river mouth. Consequently, the simulation might not wholly describe the
inundation processes and the real extent of the inundated area. These limitations aspect will be improved in future work.

Regarding the event's impact on 29 December 2018, the compound inundation occurred on a day of low precipitation. Therefore, the residents were not aware and not prepared for such a significant flood event. Even though it happened only for a short period, the economic loss was severe (Madrosid, 2018). Such loss can hopefully be avoided after the local water management installs a flood warning system built based on our proposed models.





## 5. Conclusion


In this study, a model that depicts the Kapuas River's hydrodynamic processes and its interaction with the Karimata Strait tides and wind surges has been successfully set up. We simulated the hydrodynamic processes during extreme events using the model and assessing their impact on the maximum water level along the Kapuas Kecil river. We found that wind surges over the Kapuas Kecil estuary can raise the water level up to 30 km upstream. If the backward water propagation occurs during

high river discharges, it can significantly increase the river water level and trigger an inundation over the floodplain, including the city of Pontianak.

Next, we split the main branch of the Kapuas River (Kapuas Kecil River's stream) into three regions based on the maximum water level profiles. From the river mouth to about 4 km up the river is the tidally-dominated region, where the river discharges levels do not impact the maximum water level anymore. From about 4 km to 30 km is the mixed-energy region, where the

maximum water level is influenced by the interaction between the tides and the river discharges. Then, from about 30 km upstream is the river-dominated region, where ebbs no longer impact the maximum water level. The local water management can use this zoning category in assessing and mitigating the compound flooding hazard along the riverbanks.

Lastly, as a study case, the factors that drive the inundation event over Pontianak on 29 December 2018 have been successfully investigated. The wind surge, which occurred in the estuary area, was concluded as the main trigger of the flood event. This

wind surge pushed seawater upstream and met with a high river discharge, therefore triggering a short inundation event over the floodplains, especially the city of Pontianak.

### Acknowledgment

The PhD fellowship of Joko Sampurno is provided by Indonesia Endowment Fund for Education (LPDP) under Grant No. 201712220212183. Computational resources have been provided by the supercomputing facilities of the Université catholique

de Louvain (CISM/UCL) and the Consortium des Équipements de Calcul Intensif en Fédération Wallonie Bruxelles (CÉCI) funded by the Fond de la Recherche Scientifique de Belgique (F.R.S.-FNRS) under convention 2.5020.11 and by the Walloon Region.

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

**Figure 1.** The region of interest includes the Karimata Strait, the Kapuas River and its estuary, and the city of Pontianak, whose limits are shown in green. Red dots represent the weather station's locations, and blue dots represent the tidal validation points (left: river mouth, right: Pontianak Maritime Meteorological Station observation point). Blue lines depict the river boundary where discharge is imposed in the model. Background map retrieved from (OpenStreetMap contibutors, 2017). © OpenStreetMap contributors 2017. Distributed under the Open Data
Commons Open Database License (ODbL) v1.0.





**Figure 2.** The unstructured mesh of the computational domain has a resolution of 50 m over the river and estuary, 1 km near the coastlines, and 10 km in the middle of the Karimata Strait. The mesh is composed of 206,359 triangular elements. Background map created using python library: matplotlib (Hunter, 2007).






**Figure 3.** Bathymetry map, where positive values mean under the mean sea level. Background map created using python library: matplotlib (Hunter, 2007).



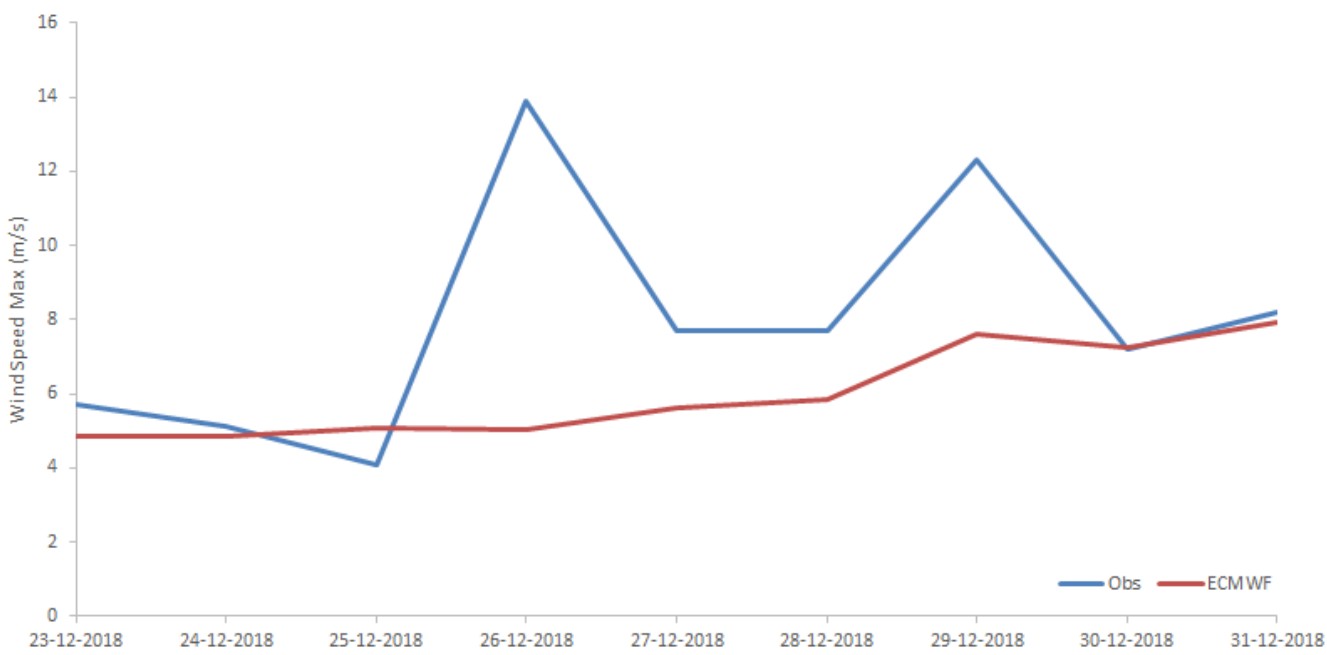

**Figure 4.** The daily maximum wind ERA5 dataset compared to the observation data at the Kapuas Kecil river mouth. It is shown that during the wind surges, the ERA5 wind product still underestimates the observation data.

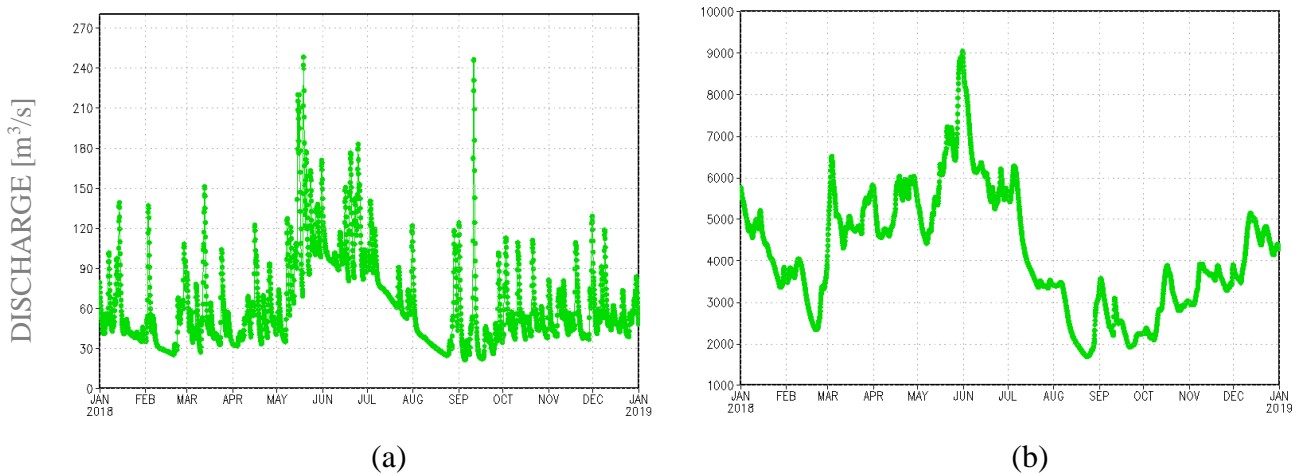

**Figure 5.** River discharge of (a) the Landak river, (b) the Kapuas river (Wu et al., 2014). Note that the Kapuas River discharge is much bigger than the Landak River discharge.




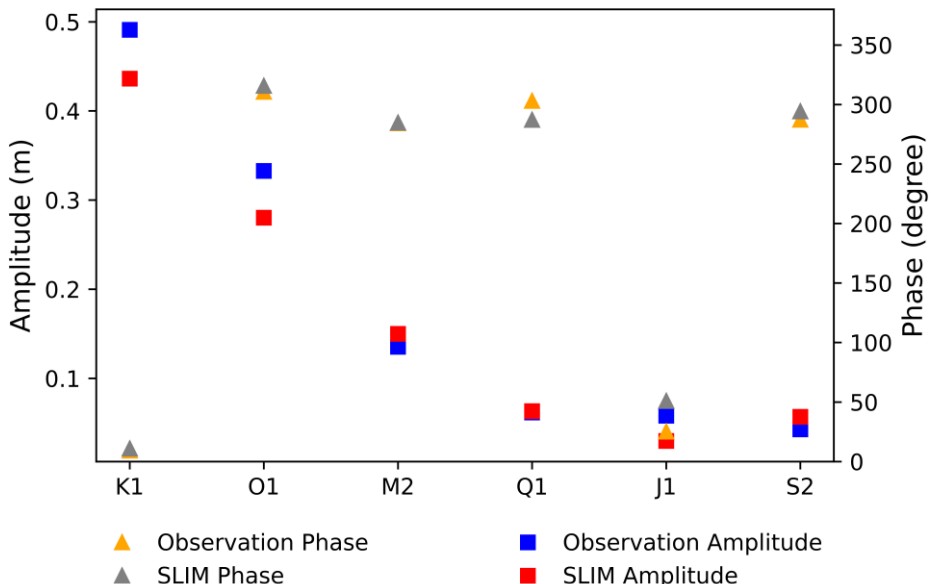

**Figure 6.** Observed and simulated tidal constituents (amplitude and phase) at the Kapuas Kecil river mouth

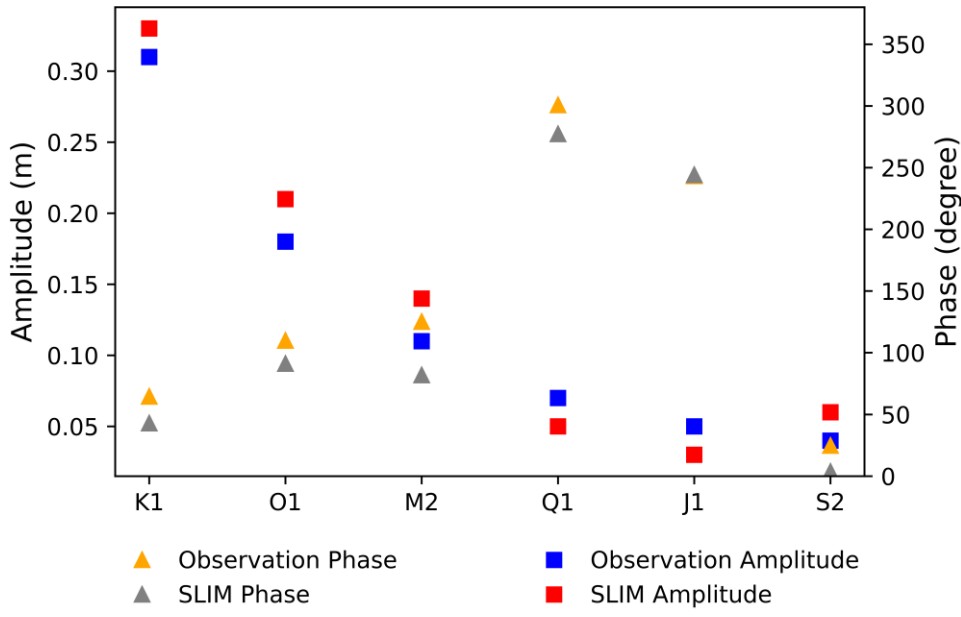


**Figure 7.** Observed and simulated tidal constituents (amplitude and phase) at the Middle of Pontianak





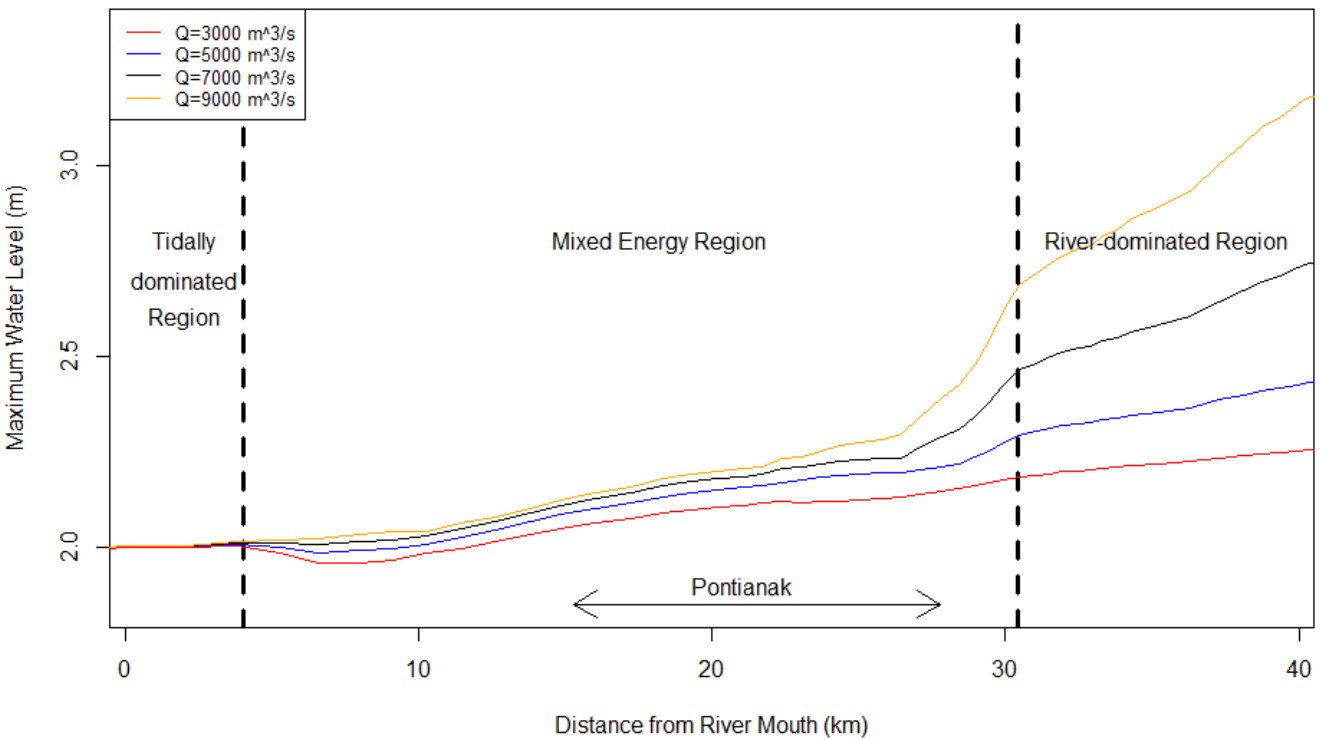


**Figure 8.** River discharge affects the maximum water level (MXWL). Black dash vertical lines are the border between the tidal-fluvial region. The arrow shows the river stream that flows inside Pontianak, where the population is the densest in the domain.





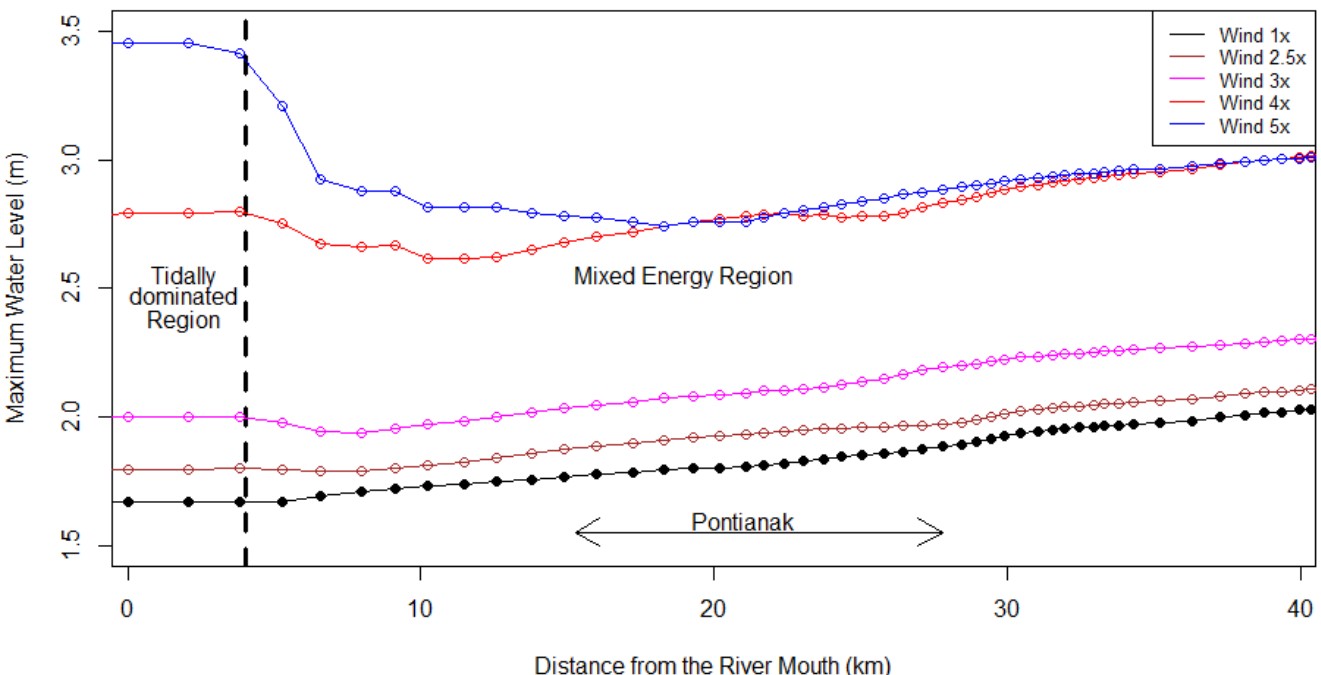

**Figure 9.** Wind surge affects the maximum water level (MXWL), where the black dash vertical line is the border between the tidal-fluvial
region. The arrow shows the river stream that flows inside Pontianak.

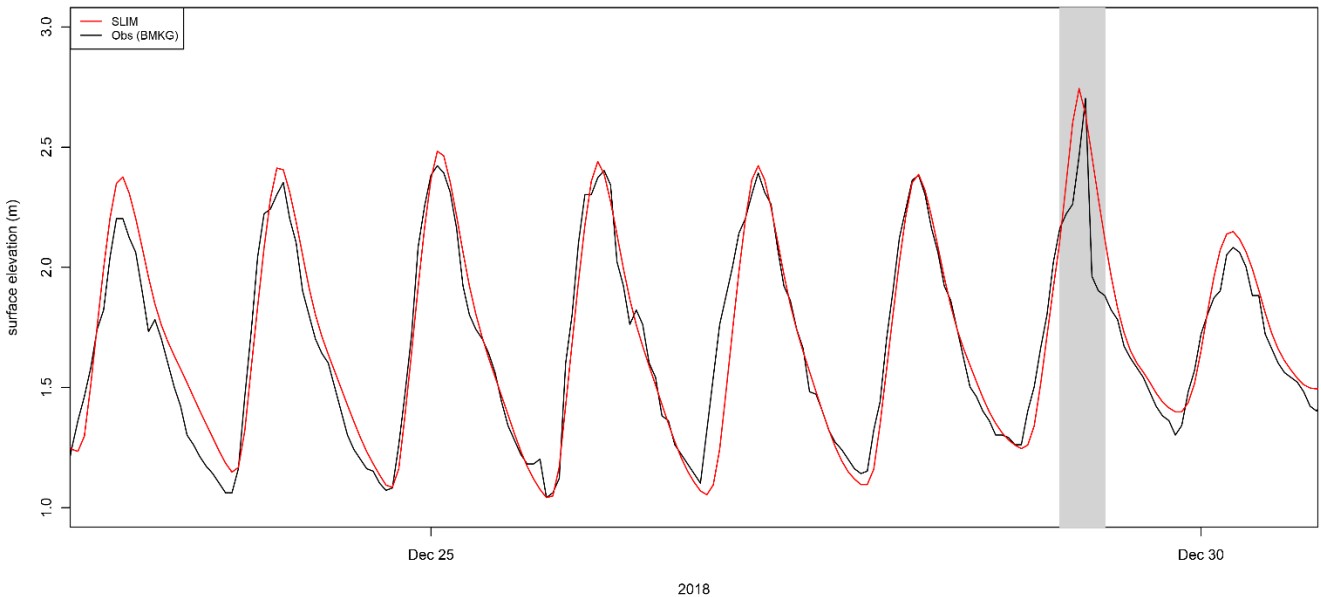

**Figure 10.** Validation of water level at Pontianak, where light grey box depicts the peak of the inundation




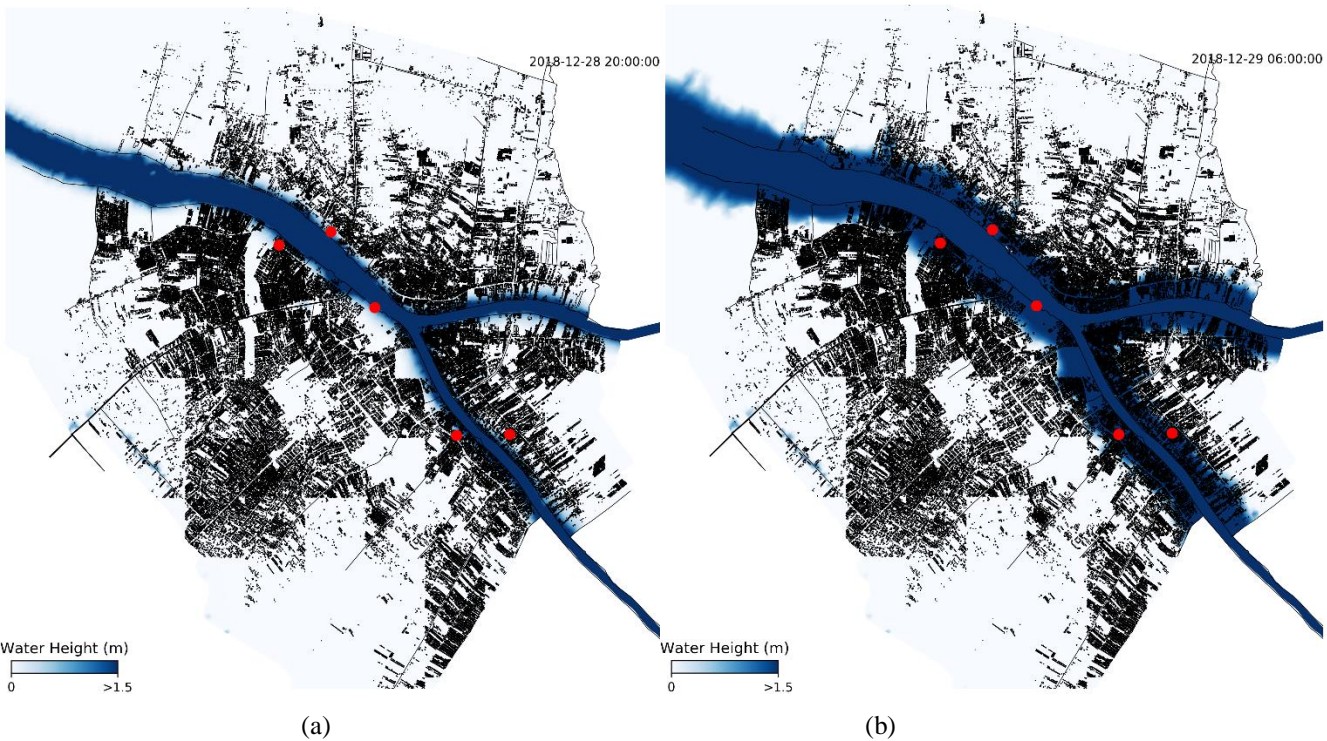

(a)                                                                 (b)

**Figure 11.** Qualitative validation map of the inundated area: (a) Before the event, (b) During the event, where the time reference is in UTC. The red dots represent places that were reported as inundated during the event. City's building map retrieved from (OpenStreetMap contibutors, 2017). © OpenStreetMap contributors 2017. Distributed under the Open Data Commons Open Database License (ODbL) v1.0.




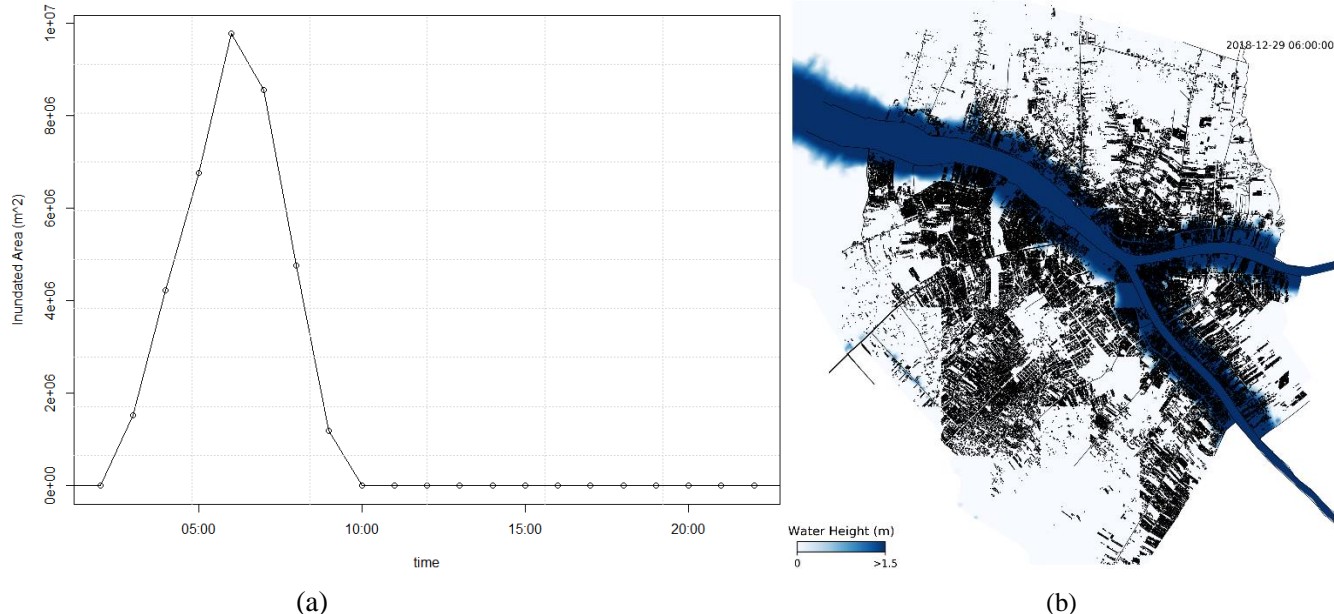

(a) (b)

**Figure 12.** The inundated area during the flood event on 29 December 2018 (a), and its depth at 06.00 UTC (b). The inundated area is determined by summing up the domain elements with a negative bathymetry (dry area), and its water level exceeds h* (dry area threshold). City's building map retrieved from (OpenStreetMap contibutors, 2017). © OpenStreetMap contributors 2017. Distributed under the Open Data Commons Open Database License (ODbL) v1.0.





**Appendix**

## A. Detail on the tidal constituents

The amplitude, phase and energy of each tidal constituent at the river mouth and in Pontianak are summarized in Table A1 and A2.

**Table A1.** Tide Constituents at The Kapuas Kecil River Mouth

| Tidal constituent | inaCOAP | | | SLIM | | | Error | | Rayleigh Criterion |
|---|---|---|---|---|---|---|---|---|---|
| | Amplitude (m) | Phase (⁰) | Percent Energy (%) | Amplitude (m) | Phase (⁰) | Percent Energy (%) | ΔAmplitude (m) | ΔPhase (⁰) | |
| K1 | 0.49 | 9.16 | 63.59 | 0.44 | 10.87 | 62.81 | 0.05 | 1.71 | 2.20 |
| O1 | 0.33 | 310.65 | 29.20 | 0.28 | 315.58 | 25.90 | 0.05 | 4.93 | 1.09 |
| M2 | 0.14 | 284.18 | 4.83 | 0.15 | 284.63 | 7.41 | 0.01 | 0.45 | 26.80 |
| Q1 | 0.06 | 303.10 | 1.00 | 0.06 | 287.09 | 1.32 | 0 | 16.01 | 24.77 |
| J1 | 0.06 | 25.15 | 0.89 | 0.03 | 50.87 | 0.29 | 0.03 | 25.72 | 1.09 |
| S2 | 0.04 | 287.16 | 0.48 | 0.06 | 294.17 | 1.07 | 0.02 | 7.01 | 2.03 |


**Table A2.** Tide Constituents at the middle of Pontianak

| Tidal constituent | Observation | | | SLIM | | | Error | | Rayleigh Criterion |
|---|---|---|---|---|---|---|---|---|---|
| | Amplitude (m) | Phase (⁰) | Percent Energy (%) | Amplitude (m) | Phase (⁰) | Percent Energy (%) | ΔAmplitude (m) | ΔPhase (⁰) | |
| K1 | 0.31 | 64.47 | 60.16 | 0.33 | 42.86 | 59.79 | 0.01 | 21.61 | 2.20 |
| O1 | 0.18 | 109.84 | 20.14 | 0.21 | 91.15 | 24.30 | 0.03 | 18.69 | 1.09 |
| M2 | 0.11 | 125.08 | 6.72 | 0.14 | 81.79 | 11.40 | 0.04 | 43.29 | 26.80 |
| Q1 | 0.07 | 300.63 | 2.64 | 0.05 | 277.21 | 1.58 | 0.01 | 23.43 | 24.77 |
| J1 | 0.05 | 243.15 | 1.78 | 0.03 | 244.13 | 0.69 | 0.02 | 0.98 | 1.09 |
| S2 | 0.04 | 24.66 | 0.81 | 0.06 | 3.62 | 1.77 | 0.02 | 21.04 | 2.03 |






**B. Local weather condition on 29 December 2018**

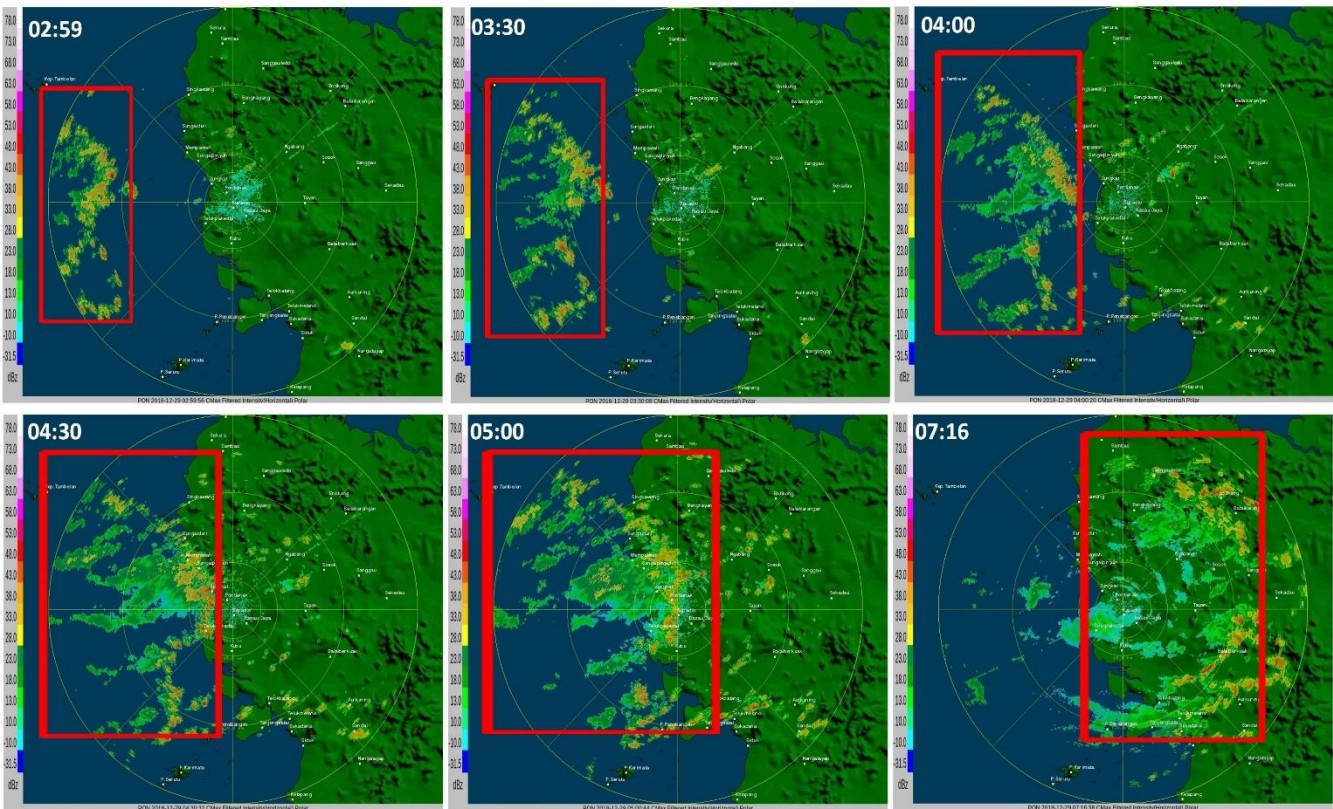

**Figure B1.** Radar data, observed by the Supadio Meteorological Station (http://kalbar.bmkg.go.id/profil/), depicts the clouds' growth and movement (in the red box) on 29 December 2018. The clouds started to grow over the ocean and are pushed by the wind to move eastward at 02.59 UTC. The clouds reach their maximum intensity at 05.00 UTC while located above the Kapuas Estuary. Furthermore, the clouds

spread over a wider area with a weakening intensity at 07.00 UTC.