# Peer review of "Modeling interactions between tides, storm surges, and river discharges in the Kapuas River delta"

_Biogeosciences, 2021_

## Referee Comment (RC2)

Dear Editor Prof. Marilaure Grégoire,

Thank you for sending me the manuscript "Modeling interactions between tides, storm surges, and river discharges in the Kapuas River delta" by Joko Sampurno et al. for review. I read the paper with great interest. The authors study a flood event caused by a storm surge in the city of Pontianak, West Kalimantan, Indonesia. The study is based on a two-dimensional numerical model and observations of wind velocity and limited gauging data in the river. This study has the potential to contribute to a better understanding, and hopefully mitigation, of flood risk in Pontianak and Indonesia. The methods are sound and the results are in agreement with previous measurements of flow and water levels in the Kapuas. However, at the moment it is focussed on one particular historic event, which is not even extreme. This is not very appealing to the general reader. This could be improved by either more systematically assessing the flood risk for Pontianak, or by providing implications for flood risk assessment in general. With that being said, the manuscript can probably be published without going that far. However, there are some technical inaccuracies, see listed below, which should be brushed out beforehand.

Kind regards,

**Major**

- The model only predicts flooding at locations near the river (c.f. line manuscript line 242), but the flooding might propagate much further into the city through the drainage channels, which seem not to be well resolved by the model. The SRTM DEM used in this study only states the surface level, the depth and hence flow velocity in the channels will be underestimated. The 30-m resolution furthermore does not horizontally resolve the channels, which are on average 5 m wide. The same applies to streets. This is aggravated by the peculiarity of SRTM to measure the highest point within each pixel.

- While the study is interesting, it does not give insight into extreme scenarios. For example, in 2013, the discharge of the Kapuas exceeded $10^4$ m$^3$/s (*Kästner et al.*, 2018). This is higher than the high flow scenario of 9000 m$^3$/s in the study, but still not overbank. It would be very insightful to provide a compound extreme value analysis of river discharge, wind and tides, and then create a flood map with likelihood of areas to be flooded in a 10, 100 and 1000 year interval, at best with incorporating the expected sea-level rise.

- The study ignores rainfall-runoff. While it is not significant for the event under study, it is relevant for the general situation in Pontianak,

as this results in flooding of large parts of the city every wet season.

**Further comments**

**2.2 Hydrodynamic model**

- State, that the model neglects the water level offset caused by the salinity gradient, and provide at least a short estimating of it (*Savenije*, 2012).

104 The equation stated are the shallow-water equations in non-conservative form, while the text says SLIM solves the conservative form. Hopefully, the latter is the case. Please correct the equations in the manuscript accordingly.

106 The Coriolis force is negligible, as it is nearly zero at the equator, where Pontianak is located. It is certainly many order of magnitudes smaller than other neglected effects, like temporal variation of roughness, salinity or secondary flows.

116 A threshold of 0.5 m seems to be too large for elements to be considered dry since flood height in the city is of the same magnitude.

**2.3 Model setup**

- I recommend extending the model domain of the Kapuas further upstream, at best until Sanngau, about 300 km from the sea. Currently, the model extends only 100 km upstream, which results in a spurious reflection of the tide, as the tide travels much farther upstream (*Kästner et al.*, 2019). The boundary of the Landak river seems also to be too close to the sea.

- Mention that the model leaves out several distributaries of the Kapuas, for example the Mendawat branch and Southern Kubu branch, and to which extend this influences the extreme water levels modelled in Pontianak.

125 Mention which data source was used to predict boundary conditions at the seaward side. (TPXO?)

134 SRTM is outdated. There is the more recent TDM global elevation map. It has also a 30 m resolution but a much higher vertical accuracy.

147 Note that roughness inferred from ADCP measurements are available for the Kapuas (*Kästner et al.*, 2018). Roughness slightly increases with the river discharge.

148 The Kapuas has a sand bed, not a "muddy river bed" (*Kästner et al.*, 2017).

174 The NSE is just the PCC applied to hydrological models. Their values should be identical and it is redundant to report both. As the reported values for the NSE and PCC are different, there seems also some inconsistency in their calculations.

- The bathymetry inset of the Kapuas Kecil shows locations with unreasonably shallow depth of just 3 m, much lower than the thalweg depth of 12 m. This might be due to bathymetry having been directly interpolated from raw data collected by *Kästner et al.* (2017). The raw data contains stretches of invalid shallow depth gauging due to faulty echo sounding which must be removed by preprocessing for obtaining a reasonably accurate bathymetry.

- Figure 5: State at which point the discharge of Wu 2014 was determined, as the Kapuas has several tributaries and distributaries along the coastal plane.

**3.1 Model validation**

186 Table A1 and A2 and Figure 6 and 7

State for which period and river discharge the tidal constituents and the goodness of fit were determined, as the constituents for the tide in Pontianak and to a limited extent at the river mouth depend on the discharge of the Kapuas.

187 Note that further data for validating the backwater curve in the upstream reach of Pontianak is available (*Kästner et al.*, 2019).

- Was a sensitivity analysis for the mouth bar depth performed? This is crucial for the backwater dynamics but probably not very accurate in the bathymetry.

**3.2 Impact of river discharge on water levels**

- State for which tidal range and date this was computed, and how this compares to the average spring tide in the Kapuas, as the impact of river discharge will depend on how strong the tide is.

- It would be informative to state which fraction of the discharge of the Kapuas is diverted into the Kapuas Kecil towards Pontianak, and to compare this with previous measurements (*Kästner and Hoitink*, 2019).

196 Figure 8: A maximum water level of 2 m at the river mouth seems to be by a factor two too large, since the maximum tidal ranges of the Kapuas is about 1.8 m. I guess this is water level with respect mean-lower-low-water (mmlw) or lowest astronomical tide (LAT). Indicate this accordingly in the caption of the figure.

**3.3 Impact of wind surges on water levels**

- Same as for 3.2, state in combination of which discharge and tidal range the wind scenarios are computed. An overview of the scenarios in a table would be meaningful.

- Discuss how the storm duration may influence the surge. Currently, only the wind force is studied.

**3.4 Case study**

- State the river discharge and the expected tidal range (without storm surge) for the date.

227 Figure 10: It would be informative to include model results (or just a fit) without wind forcing for a comparison.

**Data availability**

- Make the data, in particular for the gauging data for Pontianak, available in a public repository, as this is not yet publicly available.

**Suggested textual improvements**

10 Borneo Island → Borneo (or the island of Borneo)

42 storm surge is → storm surges are

84 The river flow ends at the Karimata Strait, creating a five-arm delta in its estuary → The river flows into the Karimata Strait through five major branches.

86 The largest distributary of the Kapuas River is the Kapuas Kecil River. → The Kapuas Kecil is the second largest distributary of the Kapuas. (Mind the name!)

87 The river starts → The river branch starts

87 20km → 20 km

88 the river flow creates a junction with the end stream of the Landak River → the Landak tributary joins the Kapuas Kecil.

89 West Borneo Province → the province of West Kalimantan (Use the current name, rather than the old colonial one.)

91 $6 \times 10^5$ → 600 000

143 Figure 4: It is preferable to plot the data as points or staircase plots, as it is discontinuous.

193 "observed data" → simulated data (since Wu et. al 2014) uses a model

197 "fully controls" → dominates → Tides are still very much important for the (maximum) water level (*Kästner et al.*, 2019)

274 the delineation of the stream zones · Unclear, explain what this means.

158 Figure 2 can be merged into 1 to save space

186 Table A2: middle of Pontianak → specify the exact coordinates

248 eastward wind → West Wind

257 Unfortunately, we failed to define → We could not define

**References**

Kästner, K., and A. J. F. Hoitink, Flow and suspended sediment division at two highly asymmetric bifurcations in a river delta: Implications for channel stability, *Journal of Geophysical Research: Earth Surface*, *124*, 2019.

Kästner, K., A. J. F. Hoitink, B. Vermeulen, T. J. Geertsema, and N. S. Ningsih, Distributary channels in the fluvial to tidal transition zone, *Journal of Geophysical Research: Earth Surface*, *3*(122), 696–710, 2017.

Kästner, K., A. J. F. Hoitink, P. J. J. F. Torfs, B. Vermeulen, N. S. Ningsih, and M. Pramulya, Prerequisites for accurate monitoring of river discharge based on fixed-location velocity measurements, *Water Resources Research*, *54*(2), 1058–1076, 2018.

Kästner, K., A. J. F. Hoitink, P. J. J. F. Torfs, E. Deleersnijder, and N. S. Ningsih, Propagation of tides along a river with a sloping bed, *Journal of Fluid Mechanics*, *872*, 39–73, 2019.

Savenije, H. H. G., *Salinity and Tides in Alluvial Estuaries, 2nd completely revised edition*, salinityandtides.com, 2012.

---

## Author Comment (AC1)

**Response to the first reviewers' comments (RC1) on the paper "*Modeling interactions between tides, storm surges, and river discharges in the Kapuas River delta*".**

We want to thank the reviewer for taking the time to review our paper. Their comment has been beneficial and helped us to improve the manuscript. In what follows, the reviewer's comments are presented in bold-italic type, our response in roman type, and modification in color in the main text.

*Page 2, 63:  can better represent.*
**Response:** The suggestion has been followed.

*Page 5: Tidal forcing: The source of the tidal signal at the open boundary is missing. How many harmonics did you use?*
**Response:** The source of the tidal signal at the open boundary is added. The number of harmonics constituents is 15, i.e., M2, S2, K1, O1, N2, P1, K2, Q1, 2N2, MF, MM, M4, MS4, MN4, and S1. We added sentences as follows in the last paragraph of page 5:

"At the open boundaries in the ocean, we prescribe tidal elevation and current of 15 harmonics from the global tides model dataset, the OSU TPXO Tide Models/TPXO9-atlas (Egbert and Erofeeva, 2002). We also retrieve global ocean circulation from HYCOM (Chassignet et al., 2007) at these boundaries".

*Wind forcing: The link/source to/of the observed wind velocity is missing. By the way, at which height are the wind data provided by ERA5 and meteorological station?*
**Response:**
The source of observed wind data is now added. Both wind data are measured and modeled at 10 m above the surface. The paragraph is updated as follows:

The wind velocity and the atmospheric pressure data are the ERA5 reanalyzes dataset obtained from the European Canter for Medium-Range Weather Forecast (ECMWF). The data have a spatial resolution of 31 km (Hersbach et al., 2020), while the temporal resolution is hourly and available at 137 vertical levels (0 to 80 km from the surface). But, here, we only selected and used the data that represented wind at the surface. Unfortunately, compared with observational data from the Stasiun Klimatologi Mempawah (http://iklim.kalbar.bmkg.go.id), measured at 10 m above the surface, there are clear difference in amplitude. The observed wind velocity is more significant than the wind velocity from ERA5 during a wind surge (Fig. 4). Therefore, in the case study, we adjust the magnitude of the wind input data (ERA5) during the wind surge event. We multiplied the wind magnitude with a ratio between both peaks (the observed and ERA5 data).

***River forcing: Not clear when the observational data are available and for which rivers. Please, clarify.***

**Response:**

The Kapuas River discharge that is available now was observed in the middle stream at Sanggau (about 284 km from the river mouth) from November 2013 to May 2015 (Kästner et al., 2018). On the other hand, the available water level data is observed by Pontianak Maritime Meteorological Station in Pontianak (20 km from the Kapuas Kecil river mouth, the second largest distributary of the Kapuas river). The data is measured hourly from 2010-2012 for only half-days (from 7 a.m. to 7 p.m.). Then, from 2012-2015 they observed the data for 15 hours (from 7 a.m. to 10 p.m.). Finally, from 2016 until now, they observed the hourly data in full days (24 times per day). Unfortunately, in our case study, in which the flood event occurred in December 2018, we don't have the observational data for the discharge, so that we replace it with the output of global discharge data, GFMS (Wu et al., 2014). We implemented sensitivity analysis to evaluate the accuracy of this GMFS's output for the Kapuas River.

***Setup: If it is possible, please, create a Table, where you describe all experiments (discharge rate, wind forcing).***

**Response:** The suggestion has been followed. We add a table in the Appendix section.

**Table A3.** Scenarios used to force the model

| Scenario | Duration of simulation | Wind Speed $(ms^{-1})$ | Wind Direction (°) | Pressure (kPa) | Discharge Kapuas $(m^3s^{-1})$ | Discharge Landak $(m^3s^{-1})$ | Tidal Range (m) |
|---|---|---|---|---|---|---|---|
| Discharge1 | 1 month | 2 - 8 | 0 – 360 | 100.5 - 101.5 | $3 \times 10^3$ | 300 | 1.8 |
| Discharge2 | 1 month | 2 - 8 | 0 – 360 | 100.5 - 101.5 | $5 \times 10^3$ | 300 | 1.8 |
| Discharge3 | 1 month | 2 - 8 | 0 – 360 | 100.5 - 101.5 | $7 \times 10^3$ | 300 | 1.8 |
| Discharge4 | 1 month | 2 – 8 | 0 – 360 | 100.5 - 101.5 | $9 \times 10^3$ | 300 | 1.8 |
| Wind_1x | 1 month | 2 – 8 | 0 – 360 | 100.5 - 101.5 | $3 \times 10^3$ | 300 | 1.8 |
| Wind_1.5x | 1 month | 3 – 12 | 0 – 360 | 100.5 - 101.5 | $3 \times 10^3$ | 300 | 1.8 |
| Wind_2x | 1 month | 4 – 16 | 0 – 360 | 100.5 - 101.5 | $3 \times 10^3$ | 300 | 1.8 |
| Wind_2.5x | 1 month | 5 – 20 | 0 – 360 | 100.5 - 101.5 | $3 \times 10^3$ | 300 | 1.8 |
| Wind_3x | 1 month | 6 - 24 | 0 – 360 | 100.5 - 101.5 | $3 \times 10^3$ | 300 | 1.8 |
| Wind_4x | 1 month | 8 - 32 | 0 – 360 | 100.5 - 101.5 | $3 \times 10^3$ | 300 | 1.8 |
| Wind_5x | 1 month | 10 - 40 | 0 – 360 | 100.5 - 101.5 | $3 \times 10^3$ | 300 | 1.8 |
| Case Study | 1 month | 2 – 21 | 0 – 360 | 100.5 - 101.5 | $3.3 \times 10^3 – 5 \times 10^3$ | 250-700 | 1.8 |

***Page 6, 165: What about P1 harmonic? It should be important for the area. Please, give some information about higher harmonics – MO3 and MK3, they would show how well your wetting/drying scheme is working.***

**Response:** The suggestion has been followed. We involved P1, MO3, and MK3 constituents in the tidal analysis. As a result, the contribution of the P1 constituent is significant, while the contribution of MO3 and MK3 constituents is relatively weak. The corresponding figure (Fig. 6 and 7), table (Table A1 and A2), and paragraph (in section 3.1 model validation) have been updated.

***Page 6, 180-189: Please, provide the coordinates of the stations.***
**Response:** The coordinates have been added.

***Page 7, 195-206: I doubt very much about MXVL analysis and zones defining procedure, especially if we consider mixed energy region. Such behavior of MXWL may signalize about larger river bed area and not about tidal impact. You can, for example, find a difference between MXVL and mean level within the tidal cycle at each location. If this difference is small, it means that the behavior of MXVL can be largely explained by a variation in river bed area. Another strategy is to run experiment with only river forcing and then find a difference between MXVL levels in experiments with tidal and river forcing and with only river forcing.***

**Response:** As suggested, we run experiments only with river forcing (without tidal forcing). We also extended our MXWL analysis area to more upstream. The result is added in Figure 8, as follows:

[Figure]

This result shows the impact of tidal and river discharge interaction on maximum water level along the river. Using the "no tide" scenario, we can see clearly that the maximum water level gradually decreases with a gentle slope in the mixed-energy zone. However, compared to the "Q=3000 m$^3$/s" scenario, we can see that the MXWL profile is more dynamic when tides are imposed in the simulation. Therefore, it supports the analysis that we mentioned in the paragraph. Based on the figure, we also modified the frontier between mix-energy and river-dominated regions to about 46 km from the river mouth.

***Page 8, 253: 'we simulated it but did not show the result here' -> 'not shown'***
**Response:** The suggestion has been followed.

*Figure 5: DISCHARGE-> Discharge. The axis font size is too small. 'Note that the Kapuas ….
discharge.' – I would remove this sentence from the caption.*
**Response:** The figure has been updated and the suggestion has been followed.

*Figure 7: The phases and amplitudes diverge larger from the observational data than they do
at the river mouth. What do you think is major reason for that?*
**Response:**
The first reason may be because the propagation of tidal waves upstream is modified by riverbanks
width convergence and depth difference (Guo et al., 2014). There is also an influence of the
backwater at the observation point within the city (Kästner et al., 2019). Therefore, the amplitude
and phase of the tide's constituents are modified. Regarding the bathymetry, the depth of the
Kapuas Kecil river mouth is very shallow, while in Pontianak, it is five-fold deeper. Next, it may
also be due to asymmetric tides as the impact of the interaction between tidal with river flow
(Parker, 1991). As we simulated, the observation point is located in the mixed energy region where
tidal and discharge interact in linear and non-linear ways. This interaction may generate other
constituents that override and decrease the amplitude and phase of constituents in the river mouth.

*Tables A1, A2: The ->the, 'Mouth'->'mouth'. Please, add P1, MO3, MK3. Please, add
coordinates of the stations.*
**Response:** The suggestion has been followed. The additional tidal constituents (P1, MO3, and
MK3) and the coordinates have been added.

*Figure 8: I think should be re-drawn, see the comments above.*
**Response:** The suggestion has been followed. Figure has been updated.

*Figure 9: Honestly, I do not understand the dynamical processes behind such a variation in
MXVL in first zone (0-4km) within different wind scenarios. It looks artificial. Can you give
some explanation? I think it would be very helpful and also add a value to the paper, if you
include the maps for each wind scenario for the considered area. You can show the MXWL
(within the tidal cycle) difference for the run with wind and tidal+river forcing and with only
tidal+river forcing.*

**Response:**
The maximum water level in this zone is the sum of the highest amplitude within the tide cycle and
the skew surge. The skew surge represents how high the sea level rises from its expected tide level
due to gradient pressure and wind stress. Since the coastal area is shallow, the stronger the wind,
the higher the skew surges possibly generated due to interaction between the tide and the surge
component (Santamaria-Aguilar and Vafeidis, 2018). The scenario we used in this simulation is
the worst scenario of wind, which blows from the ocean to the land.

[Figure]

Schematic of a skew surge (The National Oceanography Centre, 2021)

Following your suggestion, the following figure has been added to the manuscript. It depicts wind direction and magnitude over the domain for (a) scenario Wind × 1, (b) scenario wind × 2.5, (c) scenario wind × 4, and (d) scenario wind × 5.

[Figure]

We then added the MXWL profile without wind stress in Figure 10, which depicts wind surge effects on the maximum water level (MXWL) along the river.

[Figure]

*Figure 12: The axis font is hard to read, it is too small. Just a curiosity: what will be with the results, if you decrease 2 times h\*?*

**Response:**

The figure has been updated. We used h\* to define an area becomes wet or dry in our momentum equation. We have re-run the simulation using h\* = 0.1 m. Since we re-ran the model using a higher resolution of DEM and a higher mesh resolution in the case study, the city's canals can now be depicted by the model. Consequently, the inundated area extent is changing.

[Figure]

**Reference**

Chassignet, E. P., Hurlburt, H. E., Smedstad, O. M., Halliwell, G. R., Hogan, P. J., Wallcraft, A. J., Baraille, R., and Bleck, R.: The HYCOM (HYbrid Coordinate Ocean Model) data assimilative system, J. Mar. Syst., 65, 60–83, https://doi.org/10.1016/J.JMARSYS.2005.09.016, 2007.

Egbert, G. D. and Erofeeva, S. Y.: Efficient inverse modeling of barotropic ocean tides, J. Atmos. Ocean. Technol., 19, 183–204, 2002.

Guo, L., Van Der Wegen, M., Roelvin, J. A., and He, Q.: The role of river flow and tidal asymmetry on 1-D estuarine morphodynamics, J. Geophys. Res. Earth Surf., 119, 2315–2334, https://doi.org/10.1002/2014JF003110, 2014.

Hersbach, H., Bell, B., Berrisford, P., Hirahara, S., Horányi, A., Muñoz-Sabater, J., Nicolas, J., Peubey, C., Radu, R., Schepers, D., Simmons, A., Soci, C., Abdalla, S., Abellan, X., Balsamo, G., Bechtold, P., Biavati, G., Bidlot, J., Bonavita, M., Chiara, G., Dahlgren, P., Dee, D., Diamantakis, M., Dragani, R., Flemming, J., Forbes, R., Fuentes, M., Geer, A., Haimberger, L., Healy, S., Hogan, R. J., Hólm, E., Janisková, M., Keeley, S., Laloyaux, P., Lopez, P., Lupu, C., Radnoti, G., Rosnay, P., Rozum, I., Vamborg, F., Villaume, S., and Thépaut, J.: The ERA5 global reanalysis, Q. J. R. Meteorol. Soc., 146, 1999–2049, https://doi.org/10.1002/qj.3803, 2020.

Kästner, K., Hoitink, A. J. F., Torfs, P. J. J. F., Vermeulen, B., Ningsih, N. S., and Pramulya, M.: Prerequisites for Accurate Monitoring of River Discharge Based on Fixed-Location Velocity Measurements, Water Resour. Res., 54, 1058–1076, https://doi.org/10.1002/2017WR020990, 2018.

Kästner, K., Hoitink, A. J. F., Torfs, P. J. J. F., Deleersnijder, E., and Ningsih, N. S.: Propagation of tides along a river with a sloping bed, J. Fluid Mech., 872, 39–73, https://doi.org/10.1017/JFM.2019.331, 2019.

Parker, B. B.: Tidal hydrodynamics, J. Wiley, New York, 883 pp., 1991.

Santamaria-Aguilar, S. and Vafeidis, A. T.: Are Extreme Skew Surges Independent of High Water Levels in a Mixed Semidiurnal Tidal Regime?, J. Geophys. Res. Ocean., 123, 8877–8886, https://doi.org/10.1029/2018JC014282, 2018.

The National Oceanography Centre, Skew surge: https://ntslf.org/storm-surges/skew-surges, last access: 25 December 2021.

Wu, H., Adler, R. F., Tian, Y., Huffman, G. J., Li, H., and Wang, J.: Real-time global flood estimation using satellite-based precipitation and a coupled land surface and routing model, Water Resour. Res., 50, 2693–2717, https://doi.org/10.1002/2013WR014710, 2014.

---

## Author Comment (AC2)

**Response to the first reviewers' comments (RC2) on the paper "Modeling interactions between tides, storm surges, and river discharges in the Kapuas River delta".**

We want to thank the reviewer for taking the time to review our paper. Their comment has been beneficial and helped us to improve the manuscript. In what follows, the reviewer's comments are presented in italic type, our response in roman type, and modification in color in the main text.

*Major*

*• The model only predicts flooding at locations near the river (c.f. line manuscript line 242), but the flooding might propagate much further into the city through the drainage channels, which seem not to be well resolved by the model. The SRTM DEM used in this study only states the surface level, the depth and hence flow velocity in the channels will be underestimated. The 30-m resolution furthermore does not horizontally resolve the channels, which are on average 5m wide. The same applies to streets. This is aggravated by the peculiarity of SRTM to measure the highest point within each pixel.*

**Response:** The highest resolution of our mesh is 50 m; therefore, this level of detail is unreachable for our hydrodynamic model. Since our domain is very wide, using a mesh with a higher resolution than 50 m will make the computational cost expensive, and the non-hydrostatic effect will become non-negligible. As a solution, for a better inundation map extent in the case study, we re-ran a new simulation using another model HEC-RAS (2022), where the boundary conditions are set based on the output of the SLIM model. The domain of this new simulation is only enclosing the city of Pontianak. This new simulation uses 10 m mesh resolution and replaces the SRTM using a higher resolution DEM from DEMNAS (https://tanahair.indonesia.go.id/demnas/). The new DEM map has a 0.27-arcsecond (about 8.3 m) resolution. Therefore, using this strategy, we could evaluate the flows inside the canals that drive inundation over locations far from the river banks. We then modified the flood extent map as follows:

[Figure]

• *While the study is interesting, it does not give insight into extreme scenarios. For example, in 2013, the discharge of the Kapuas exceeded $10^4$ m3/s (Kastner et al., 2018). This is higher than the high ow scenario of 9000 m3/s in the study, but still not overbank. It would be very insightful to provide a compound extreme value analysis of river discharge, wind and tides, and then create a flood map with likelihood of areas to be flooded in a 10, 100, and 1000 years interval, at best with incorporating the expected sea-level rise.*

**Response:** As we show in the river discharge scenario, increasing the discharge until 9000 m3/s doesn't significantly increase the maximum water level (MXWL) at Pontianak. MXWL in this river zone is influenced by the interaction of tide-surge and discharge altogether. It may explain why the only 10000 $m^3$/s discharge doesn't drive inundation over the city. It needs other factors, such as storm surge or excessive rainfall, to drive inundation over the city. Regarding flood frequency analysis which considers sea-level rise due to climate change, it will be investigated in future studies.

• *The study ignores rainfall-runoff. While it is not significant for the event under study, it is relevant for the general situation in Pontianak, as this results in flooding of large parts of the city every wet season.*
**Response:** As we mentioned in the discussion part, it is part of the limitation of this study. Here, we only consider the interaction between river discharges, tides, and wind surge and how their interaction drive inundation over the city. As presented in the case study, the model focuses on inundation events during low rainfall over the city; therefore, the rainfall impact is not taken into account. We will investigate the rainfall-runoff impact in future studies.

**2.2 Hydrodynamic model**

• ***State,*** *that the model neglects the water level offset caused by the salinity gradient, and provide at least a short estimating of it (Savenije, 2012).*
**Response:** Since our model is barotropic, we do not represent salinity. We can therefore not estimate its value from the model results. It could possibly be estimated by using the outputs of HYCOM, but since we don't know if the discharge of the Kapuas river is correctly imposed in HYCOM, there is no guarantee that the freshwater plume (and hence the salinity gradient) will be correct.

***104*** *The equation stated are the shallow-water equations in non-conservative form, while the text says SLIM solves the conservative form. Hopefully, the latter is the case. Please correct the equations in the manuscript accordingly.*
**Response:** The equations are already in conservative form. As mentioned in the manuscript, $U = H\overline{u}$ is the horizontal transport (and not velocity).

*106 The Coriolis force is negligible, as it is nearly zero at the equator, where Pontianak is located. It is certainly many orders of magnitudes smaller than other neglected effects, like temporal variation of roughness, salinity or secondary flows.*

**Response:** We include a large portion of the Karimata Strait in our domain, which the latitude extent from -2.8 degrees to 1.8 degrees. This large portion of the Strait is included because we want to evaluate how wind surge, which may occur offshore, impact MXWL along the Kapuas Kecil. The Coriolis force is indeed not the main force during the dynamics. However, as it is cheap to compute, we did not remove it from the model even if its contribution is very limited.

*116 A threshold of 0.5 m seems to be too large for elements to be considered dry since flood height in the city is of the same magnitude.*

**Response:** We re-ran the simulation using h\* = 0.1 m as the threshold and used a higher resolution of DEM and a higher mesh resolution in the case study; therefore, the inundated area extent is now changing.

**2.3 Model setup**

*• I recommend extending the model domain of the Kapuas further upstream, at best until Sanngau, about 300 km from the sea. Currently, the model extends only 100 km upstream, which results in a spurious reflection of the tide, as the tide travels much farther upstream (Kastner et al., 2019). The boundary of the Landak river seems also to be too close to the sea.*

**Response:**

Many small tributaries are joining, and some distributaries are leaving the Kapuas stream between the current boundary condition and Sanggau. Unfortunately, we do not have these tributaries/distributaries' discharge data. While, based on GFMS output (calculated using real-time TRMM Multi-satellite Precipitation Analysis and Global Precipitation Measurement), there are different discharges between the current boundary and discharge retrieved in Sanggau (see the following figure). So, if we put boundary conditions at Sanggau, it will neglect the contribution of these tributaries and distributaries. Therefore, we decided to put the Kapuas boundary condition 10 km upstream before the Kapuas Kecil branch starting point as a reasonable approach.

[Figure]

The Kapuas River discharge is retrieved from GFMS during December 2021 at (a) Sanggau with discharge range 5600 m³/s to 6900 m³/s, and (b) Current boundary condition, located 10 km upstream before the Kapuas Kecil branch starting point, with discharge range 6150 m³/s to 7250 m³/s.

Regarding the spurious reflection of the tides, we still have no idea how to prove it. However, our focus in this study is evaluating whether tides still dominantly control the maximum water level to drive inundation over the floodplain in certain areas along the Kapuas Kecil river. Therefore, our result suggests that even though tides travel much further upstream, it doesn't dominantly impact inundation anymore after 46 km further.

*• Mention that the model leaves out several distributaries of the Kapuas, for example the Mendawat branch and Southern Kubu branch, and to which extend this influences the extreme water levels modelled in Pontianak.*
**Response:** The suggestion has been followed. We added sentences as follows in this section:
"Since we focused on evaluating the impact of tide-surge-discharge interaction on extreme water levels along the Kapuas Kecil branch (particularly in Pontianak), we leave out several distributaries that may not significantly influence that dynamics, such as the Southern Kubu branch."

*125 Mention which data source was used to predict boundary conditions at the seaward side. (TPXO?)*
**Response:** The suggestion has been followed. We added sentences as follows in this section:

"At the open boundaries in the ocean, we prescribe tidal elevation and current of 15 harmonics from the global tides model dataset, the OSU TPXO Tide Models/TPXO9-atlas (Egbert and Erofeeva, 2002). We also retrieve global ocean circulation from HYCOM (Chassignet et al., 2007) at these boundaries".

*134 SRTM is outdated. There is the more recent TDM global elevation map. It has also a 30m resolution but a much higher vertical accuracy.*
**Response:**
We replaced the SRTM map with a higher resolution map from DEMNAS (https://tanahair.indonesia.go.id/demnas/) to obtain a better inundation map over the city. The new DEM has spatial resolution 0.27-arcsecond (about 8.3 m).

*147 Note that roughness inferred from ADCP measurements are available for the Kapuas (Kastner et al., 2018). Roughness slightly increases with the river discharge.*
**Response:**
The suggestion has been followed. We re-ran the simulation using the new roughness coefficient, which defined based on (Kästner et al., 2018). However, the result is not too different.

*148 The Kapuas has a sand bed, not a "muddy river bed" (Kastner et al., 2017).*
**Response:** The suggestion has been followed. We modified the text accordingly.

***174** The NSE is just the PCC applied to hydrological models. Their values should be identical and it is redundant to report both. As the reported values for the NSE and PCC are different, there seems also some inconsistency in their calculations.*

**Response:** We calculated PCC and NSE parameters using the library `HydroGof` in R (Zambrano-Bigiarini, 2020). the results show that both values are consistently different. Other studies (Kumar et al., 2020; Zhou et al., 2019) also reported that they implement both PCC and NSE as the goodness of fit in their work, which the result of both values is also different. So, we couldn't find the inconsistency in our calculation. However, we decided to remove the PCC and kept the NSE (from the library `HydroGof` in R) as the goodness of fit coefficient, as NSE is more common to assess the predictive skill of hydrological models.

*• The bathymetry inset of the Kapuas Kecil shows locations with unreasonably shallow depth of just 3 m, much lower than the thalweg depth of 12 m. This might be due to bathymetry having been directly interpolated from raw data collected by Kastner et al. (2017). The raw data contains stretches of invalid shallow depth gauging due to faulty echo sounding which must be removed by preprocessing for obtaining a reasonably accurate bathymetry.*

**Response:**

Since we used the Wetting-Drying algorithm, we used both positive (underwater) and negative bathymetries (dry area from DEM) altogether. So, the bathymetry values in the map range from -3 m (upper mean sea-level) to 100 m (under mean sea-level). Minus values represent the dry area and vice versa. The blue areas represent the dry land. For the locations around the river mouth, we compared our bathymetry with the bathymetry map from Garmin (see Figure below). Overall, we found that the river mouth bathymetry is similar (which is shallow).

[Figure]

The bathymetry map around the Kapuas Kecil river mouth from Garmin
(https://webapp.navionics.com/?lang=en#boating@10&key=kpIw%7BmyS)

Regarding the Thalweg depth, there is a narrow-deeper flow path for the shipping route, which cross the river mouth (see figure above). However, as we saw in our field trip, ships still cannot pass through this flow path during low tide sessions due to its shallowness. In addition, the bathymetry should also be smoothed in our simulation to make it run stable. The smoothing process is based on mesh points, where, unfortunately, the highest mesh element cannot capture whole of this narrow-deeper path. In some parts, the smoothed map overlays this narrow path using

interpolated depth among its surrounding areas. This is the shallow area, which may you saw on the inset map.

• *Figure 5: State at which point the discharge of Wu 2014 was determined, as the Kapuas has several tributaries and distributaries along the coastal plane.*
**Response:** The suggestion has been followed. We modified the figure's caption as follows:
"River discharge of (a) the Landak (prescribed at coordinate 0.0282S, 109.445E) and (b) the Kapuas (prescribed at coordinate 0.3623S, 109.6394E) retrieved from Wu et al., (2014)."

**3.1 Model validation**

*186 Table A1 and A2 and Figure 6 and 7 State for which period and river discharge the tidal constituents and the goodness of fit were determined, as the constituents for the tide in Pontianak and to a limited extent at the river mouth depend on the discharge of the Kapuas.*
**Response:** The data is observed and simulated for one month in December 2018. The simulation retrieved river discharge from GFMS (Wu et al., 2014) and tide from TPXO (Egbert and Erofeeva, 2002). This information has been added in the caption of the figures and tables.

*187 Note that further data for validating the backwater curve in the upstream reach of Pontianak is available (Kastner et al., 2019).*
**Response:** We thank the reviewer for this suggestion. We will try to do that in a future study. Since we used the maximum water level (MXWL) instead of the mean water level to evaluate the tide-surge-discharge impacts, we didn't analyze the backwater curve in this study. This MXWL includes the skew surge as the impact of the wind surge in our wind scenarios.

[Figure]

Schematic of a skew surge (The National Oceanography Centre, 2021)

*• Was a sensitivity analysis for the mouth bar depth performed? This is crucial for the backwater dynamics but probably not very accurate in the bathymetry.*

**Response:** Since our computational domain already covers both river bodies (wet area) and floodplain areas over riverbanks (dry areas), we didn't perform sensitivity analysis for the mouth bar depth.

**3.2 Impact of river discharge on water levels**

*• State for which tidal range and date this was computed, and how this compares to the average spring tide in the Kapuas, as the impact of river discharge will depend on how strong the tide is.*

**Response:** The suggestion has been followed. We modified the second paragraph as follows:

"At the same time, the Landak River upstream discharge is set to 300 m3/s for all scenarios, and the tide used in the ocean part is retrieved from TPXO for December 2018 with a tidal range of 1.8 m at the river mouth."

*• It would be informative to state which fraction of the discharge of the Kapuas is diverted into the Kapuas Kecil towards Pontianak, and to compare this with previous measurements (Kastner and Hoitink, 2019).*

**Response:** We calculated the portion discharge of Kapuas, which diverted through the Kapuas Kecil branch during December 2018, is about 16% (see Figure below), while 84% continue to flow through the Kapuas Besar. This result is close to the observational data, which stated that 17% of the Kapuas discharge flows into the Kapuas Kecil, and 83% continue along the Kapuas Besar (Kästner and Hoitink, 2019). We added the information to this section.

[Figure]

**196** *Figure 8: A maximum water level of 2m at the river mouth seems to be by a factor two too large, since the maximum tidal ranges of the Kapuas is about 1.8 m. I guess this is water level with respect meanlower-low-water (mmlw) or lowest astronomical tide (LAT). Indicate this accordingly in the caption of the figure.*

**Response:** The suggestion has been followed. The figure's caption has been modified.

**3.3 Impact of wind surges on water levels**

*• Same as for 3.2, state in combination of which discharge and tidal range the wind scenarios are computed. An overview of the scenarios in a table would be meaningful.*

**Response:** We add a table that describe our scenario in the Appendix section.

**Table A3.** Scenarios used to force the model

| Scenario | Duration of simulation | Wind Speed $(ms^{-1})$ | Wind Direction (°) | Pressure (kPa) | Discharge Kapuas $(m^3s^{-1})$ | Discharge Landak $(m^3s^{-1})$ | Tidal Range (m) |
|---|---|---|---|---|---|---|---|
| Discharge1 | 1 month | 2 - 8 | 0 – 360 | 100.5 - 101.5 | $3 \times 10^3$ | 300 | 1.8 |
| Discharge2 | 1 month | 2 - 8 | 0 – 360 | 100.5 - 101.5 | $5 \times 10^3$ | 300 | 1.8 |
| Discharge3 | 1 month | 2 - 8 | 0 – 360 | 100.5 - 101.5 | $7 \times 10^3$ | 300 | 1.8 |
| Discharge4 | 1 month | 2 – 8 | 0 – 360 | 100.5 - 101.5 | $9 \times 10^3$ | 300 | 1.8 |
| Wind_1x | 1 month | 2 – 8 | 0 – 360 | 100.5 - 101.5 | $3 \times 10^3$ | 300 | 1.8 |
| Wind_1.5x | 1 month | 3 – 12 | 0 – 360 | 100.5 - 101.5 | $3 \times 10^3$ | 300 | 1.8 |
| Wind_2x | 1 month | 4 – 16 | 0 – 360 | 100.5 - 101.5 | $3 \times 10^3$ | 300 | 1.8 |
| Wind_2.5x | 1 month | 5 – 20 | 0 – 360 | 100.5 - 101.5 | $3 \times 10^3$ | 300 | 1.8 |
| Wind_3x | 1 month | 6 - 24 | 0 – 360 | 100.5 - 101.5 | $3 \times 10^3$ | 300 | 1.8 |
| Wind_4x | 1 month | 8 - 32 | 0 – 360 | 100.5 - 101.5 | $3 \times 10^3$ | 300 | 1.8 |
| Wind_5x | 1 month | 10 - 40 | 0 – 360 | 100.5 - 101.5 | $3 \times 10^3$ | 300 | 1.8 |
| Case Study | 1 month | 2 – 21 | 0 – 360 | 100.5 - 101.5 | $3.3 \times 10^3 – 5 \times 10^3$ | 250-700 | 1.8 |

*• Discuss how the storm duration may influence the surge. Currently, only the wind force is studied.*

**Response:** The suggestion has been followed. We added wind surge duration impacts in this section. We added following figure:

[Figure]

And added a paragraph in this section, as follows:

"Besides the wind velocity, the flood duration and flood extent along the river are also influenced by the storm duration (Höffken et al., 2020). Figure 10 shows that the impact of the wind surge

duration on the maximum water level is not significant, but the flood event occurred longer. Backwater comes from the river mouth upstream and stays longer inland before flowing back to the ocean."

**3.4 Case study**
• *State the river discharge and the expected tidal range (without storm surge) for the date.*
**Response:** The suggestion has been followed. We added following sentences:
"We simulated the hydrodynamical process without and with storm surge scenarios when the expected tidal range during the event is 1.8 m. Since there was no observed discharge upstream during the date, we imposed the river discharge retrieved from GFMS for Kapuas and Landak rivers (Table A3)."

***227** Figure 10: It would be informative to include model results (or just a fit) without wind forcing for a comparison.*
**Response:** The suggestion has been followed. We updated Figure 10 by adding the water level dynamics in Pontianak, which simulated without wind forcing. As seen in the figure, based on the "without wind forcing" scenario, the peak water level within the tidal cycle during the event day should be lower than the peak in the previous day. Therefore, the storm is responsible for a 30 cm increase in the water level (grey box in the figure). In addition, the water level surge happened after the tidal cycle passed its peak and in the move to decrease.

[Figure]

**Data availability**
• *Make the data, in particular for the gauging data for Pontianak, available in a public repository, as this is not yet publicly available.*
**Response:**
We have already published the gauged water level in Pontianak for 2018, which we used in the case study, at https://doi.org/10.5281/zenodo.5809647.

**Suggested textual improvements**

*10 Borneo Island -> Borneo (or the island of Borneo)* **Response:** The suggestion has been followed.

*42 storm surge is -> storm surges are,* **Response:** The suggestion has been followed.

*84 The river flow ends at the Karimata Strait, creating a five-arm delta in its estuary -> The river flows into the Karimata Strait through five major branches.*
**Response:** The suggestion has been followed.

*86 The largest distributary of the Kapuas River is the Kapuas Kecil River. -> The Kapuas Kecil is the second largest distributary of the Kapuas. (Mind the name!)*
**Response:** The suggestion has been followed.

*87 The river starts ->The river branch starts,* **Response:** The suggestion has been followed.

*87 20km-> 20 km,* **Response:** The suggestion has been followed.

*88 the river flow creates a junction with the end stream of the Landak River-> the Landak tributary joins the Kapuas Kecil.*
**Response:** The suggestion has been followed.

*89 West Borneo Province -> the province of West Kalimantan (Use the current name, rather than the old colonial one.)*
**Response:** The suggestion has been followed.

*91 $6\times10^5$ -> 600 000* **Response:** The suggestion has been followed.

*143 Figure 4: It is preferable to plot the data as points or staircase plots, as it is discontinuous.*
**Response:** The suggestion has been followed.

*193 "observed data" -> simulated data (since Wu et. al 2014) uses a model*
**Response:** The suggestion has been followed.

*197 "fully controls" -> dominates -> Tides are still very much important for the (maximum) water level (Kastner et al., 2019)*
**Response:** The suggestion has been followed.

*274 the delineation of the stream zones -> Unclear, explain what this means.*
**Response:** We updated the sentence become as follows:
Therefore, the delineation of the compound flooding risk zones based on the MXWL proposed for the Kapuas Kecil River needs further investigation in the future.
*158 Figure 2 can be merged into 1 to save space*
**Response:** The suggestion has been followed.

*186 Table A2: middle of Pontianak -→specify the exact coordinates*
**Response:** The coordinates has been added.

*248* eastward wind -> West Wind
**Response:** The suggestion has been followed.

*257* Unfortunately, we failed to define -> We could not define
**Response:** The suggestion has been followed.

**Reference**

Chassignet, E. P., Hurlburt, H. E., Smedstad, O. M., Halliwell, G. R., Hogan, P. J., Wallcraft, A. J., Baraille, R., and Bleck, R.: The HYCOM (HYbrid Coordinate Ocean Model) data assimilative system, J. Mar. Syst., 65, 60–83, https://doi.org/10.1016/J.JMARSYS.2005.09.016, 2007.

Egbert, G. D. and Erofeeva, S. Y.: Efficient inverse modeling of barotropic ocean tides, J. Atmos. Ocean. Technol., 19, 183–204, 2002.

Höffken, J., Vafeidis, A. T., MacPherson, L. R., and Dangendorf, S.: Effects of the Temporal Variability of Storm Surges on Coastal Flooding, Front. Mar. Sci., 7, 98, https://doi.org/10.3389/FMARS.2020.00098/BIBTEX, 2020.

Kästner, K. and Hoitink, A. J. F.: Flow and Suspended Sediment Division at Two Highly Asymmetric Bifurcations in a River Delta: Implications for Channel Stability, J. Geophys. Res. Earth Surf., 124, 2358–2380, https://doi.org/10.1029/2018JF004994, 2019.

Kästner, K., Hoitink, A. J. F., Torfs, P. J. J. F., Vermeulen, B., Ningsih, N. S., and Pramulya, M.: Prerequisites for Accurate Monitoring of River Discharge Based on Fixed-Location Velocity Measurements, Water Resour. Res., 54, 1058–1076, https://doi.org/10.1002/2017WR020990, 2018.

Kumar, M., Kumari, A., Kushwaha, D. P., Kumar, P., Malik, A., Ali, R., and Kuriqi, A.: Estimation of Daily Stage–Discharge Relationship by Using Data-Driven Techniques of a Perennial River, India, Sustain. 2020, Vol. 12, Page 7877, 12, 7877, https://doi.org/10.3390/SU12197877, 2020.

Savenije, H. H. G.: Salinity and Tides in Alluvial Estuaries, Second Com., Delft, 2012.

The National Oceanography Centre, Skew surge: https://ntslf.org/storm-surges/skew-surges, last access: 25 December 2021.

HEC-RAS: https://www.hec.usace.army.mil/software/hec-ras/, last access: 8 January 2022.

Wu, H., Adler, R. F., Tian, Y., Huffman, G. J., Li, H., and Wang, J.: Real-time global flood estimation using satellite-based precipitation and a coupled land surface and routing model, Water Resour. Res., 50, 2693–2717, https://doi.org/10.1002/2013WR014710, 2014.

Zambrano-Bigiarini, M.: Goodness-of-Fit Functions for Comparison of Simulated and Observed Hydrological Time Series [R package hydroGOF version 0.4-0], 2020.

Zhou, L., Fok, H. S., Ma, Z., and Chen, Q.: Upstream Remotely-Sensed Hydrological Variables and Their Standardization for Surface Runoff Reconstruction and Estimation of the Entire Mekong River Basin, Remote Sens. 2019, Vol. 11, Page 1064, 11, 1064, https://doi.org/10.3390/RS11091064, 2019.

---

## Referee Report (RR1)

The authors have improved the manuscript and addressed nearly all comments.

However, some minor issues should be still considered before the final publication:

**General issues:**

Pg 8, 215:  'Then, from this point to a point located at about 4 km from the river mouth, all MXWL profiles start to drop, indicating that there is a compounding effect of the discharge and the tide.'- I think it is not correct sentence. The drop itself does not mean the compounding effect. We can trace it also in Fig. 7, where the case without tidal forcing is shown. The reason of this drop lies in the topography details and riverbed area change. In some rivers, you can trace several such dropping of the level moving from upstream to the delta without any compounding effect. (Of course, these topography details can be also a reason that the tidal signal reaches the point). Please, put attention to this point and give an explanation of the drop in MXWL.

Pg 8, 222-223: 'This stream part, where both the river discharge and the tide control the MXWL, is called a mixed-energy region'

The definitions of the River-dominated and Mixed regions are not clear and contradictory to what we see in Figure 7.  If we consider 2 cases with tidal forcing off and on and discharge equaled to 3000 m^3/s, we clearly see, that the presence of tides influences the MXWL till 70 km! from River mouth (Figure 7). So based on this definition, the mixed region starts nearly from the border, where river source is prescribed. From another point of view, you can say, that the MXWL is defined by the river until MXVL reaches ~2m. Based on this definition you will get an extended River-dominated area compared to present in the manuscript, and this area will be getting larger increasing the discharge. There can be more definitions, please, clarify yours.

**Technical:**

Pg 3, 58-63:  Can be shortened to the point:  The area of interest represents well-mixed and relatively shallow water body. Therefore, we applied 2D barotropic solution to reduce computational costs.

Pg 7, 172: please, remove 'with satisfying frequencies' (you have already identified that in the beginning of the sentence, can be interpreted wrong).

Figure 8: Please, leave only one subpanel, e.g., (a).

Table A3: The details, which are the same for all experiments, can be identified in the Figure caption.

Figure 3: The Figure is nice, but the font is really small compared to other pictures.

---

## Referee Report (RR2)

Dear Editor Marilaure Grégoire,

Thank you for sending me the revised manuscript "Modeling interactions between tides, storm surges, and river discharges in the Kapuas River delta" by Sampurno et al. for review. I enjoyed reading the manuscript very much. The authors study with a detailed numerical model a compound flood induced by a storm surge in the city of Pontianak. The authors have responded to the review comments and improved their manuscript accordingly. In particular flooding in the city through the drainage channels is now reproduced. The study addresses a pressing issue facing many cities in Southeast Asia and beyond. Therefore I recommend the manuscript for publication. I give some short comments below for finalizing the manuscript.

Kind regards,

**Minor comments**

Nested model: This approach is interesting. Is there a particular reason why the mesh of the large scale SLIM model has not simply been refined at the drainage channels? Since SLIM allows for local mesh refinement and uses implicit time-stepping, I would not expect a large penalty on the runtime. Even with two models, why was HEC-RAS chosen over SLIM for the nested model?

Upstream boundary: The authors explain that they placed the upstream boundary near Terentang about 100 km upstream from the sea, to avoid missing discharge from tributaries (mostly Tayan and Meliau) downstream of the head of tides at about km 300, near Sanggau. While this approach indeed reproduces the river discharge at the inflow boundary, it cuts the tidal prism, and thereby reflects the tidal wave and reduces the tidal discharge. The figure below shows the tidal discharge estimated with the theory of tides (*Hill and Souza*, 2006; *Kästner et al.*, 2019). Truncating the domain as in the numerical model reduces the tidal discharge in the Kapuas Besar branch by 50% and increase the tidal discharge in the Kapuas Kecil branch by 30%. To get both the river and tidal discharge right the boundary could be moved to Sanggau while the inflow is set to the discharge measured at Terentang.

[Figure]

Bathymetry: My comment on erroneously shallow cross-sections in the original manuscript was not clear. What I mean is not the mouth bar but that the raw bathymetry data of (*Kästner et al.*, 2017) erroneously contains shallow cross-sections between Pontianak and the upstream bifurcation. This is due to glitches of the echo sounder used for the measurement. The SLIM model results show jumps in surface elevation at km 30 and 45. This seems physically implausible and is probably due to backwater caused by erroneous constrictions of the cross-section. I suggest verifying this and if applicable, filtering the bathymetry along-channel.

Terminology: I agree with the first reviewer who commented that the adopted zone-terminology is somewhat confusing. The terms tidal energy and maximum water level are used interchangeably throughout the manuscript. However, there is no direct correspondence between the maximum water level and the (kinetic) energy. The maximum water level is a combination of the tidal amplitude and tidally averaged water level. The effect of tides on the mean water level is largest upstream of the point where most of the tidal energy has already been dissipated, as it integrates along channel, while the tidal amplitude decreases gradually along channel. The storm surge, furthermore, contributes an important part to the energy budget. Therefore, I recommend referring to water levels throughout the manuscript, and avoiding the term "energy".

**Typography**

19 could divide → divide

105 For the wind shear stress a surface roughness is required, similar to $c_d$ for the bed shear stress. What value was chosen?

188,189 new mesh → second mesh

194 0.09m → 0.09 m

197 semidiurnal components explain the rest → there is no rest (90.69 + 9.31 = 100)

231 will drop → drops

234 leads to a reduction in the water levels → reduced the water level

236 not too significant → not significant

242 State in here that the reference for the 2.8 m water level is the lowest astronomical tide (LAT) and that the 2.8 m correspond roughly to 1.8 m above mean sea level and 0.7 m above highest astronomical tide (HAT). State also the river discharge for this day.

245 Please state the Kapuas discharge and tidal range (without storm surge) for that day!

251 top → high water level?

263 Landak river streams → Landak River

277 the wind velocity less than 9 m/s or more than 24 m/s, it does not → wind velocities less than 9 m/s or more than 24 m/s do not

281 zone border → boundary

281 mix-energy → mixed-energy

282 border → boundary

293 from the river mouth to the upstream → upstream from the river mouth

293 was coincidentally met with a high river discharge → I would call this more an intermediate discharge, as it seems to be less than 1/2 of annual peak discharge of the river.

321 where ebbs no longer impact → where tides no longer impact

Figure 7 It would be insightful to complement this figure with an along channel plot of tidal range and tidally averaged water level.

Figure 13 Limit the colourmap of water depth between 0 m and 2 m, to better distinguish flooding in the city.

**References**

Hill, A. E., and A. J. Souza, Tidal dynamics in channels: 2. Complex channel networks, *Journal of Geophysical Research: Oceans*, *111*(C11), 2006.

Kästner, K., A. J. F. Hoitink, B. Vermeulen, T. J. Geertsema, and N. S. Ningsih, Distributary channels in the fluvial to tidal transition zone, *Journal of Geophysical Research: Earth Surface*, *3*(122), 696–710, 2017.

Kästner, K., A. J. F. Hoitink, P. J. J. F. Torfs, E. Deleersnijder, and N. S. Ningsih, Propagation of tides along a river with a sloping bed, *Journal of Fluid Mechanics*, *872*, 39–73, 2019.

---

## Author Response (AR2)

**Response to the second comments of the first reviewer (RC1)**

We want to thank the reviewer again for having taken the time to review again our paper. His/her comments were really useful to help us further improve the manuscript.

**General issues:**

**Pg 8, 215:** 'Then, from this point to a point located at about 4 km from the river mouth, all MXWL profiles start to drop, indicating that there is a compounding effect of the discharge and the tide.'- I think it is not correct sentence. The drop itself does not mean the compounding effect. We can trace it also in Fig. 7, where the case without tidal forcing is shown. The reason of this drop lies in the topography details and riverbed area change. In some rivers, you can trace several such dropping of the level moving from upstream to the delta without any compounding effect. (Of course, these topography details can be also a reason that the tidal signal reaches the point). Please, put attention to this point and give an explanation of the drop in MXWL.

**Response:** *We agree with the reviewer. The drop could indeed be due to geometric changes in the river body (strong meandering) and rapid changes in the river depth. We therefore removed this statement.*

*We also re-ran the simulation with a further upstream boundary (as suggested by Reviewer 2). As a result, the MXWL profile changed, where the maximum water level at the Kapuas Kecil river mouth is equal to the tidal range, but this drop pattern is consistent. We also more closely analyzed the river bathymetry and topography in that area. In this river part, we found that the bathymetry drops from 11 m to 16 m at TelokKumpai, while at Baru, the river width is narrower (see Figure below). So, this drop pattern could be due to the rapid change in river depth and width.*

[Figure]

*Source: https://webapp.navionics.com/#boating@10&key=%60wT%7Dk%7BzS*

**Pg 8, 222-223:** 'This stream part, where both the river discharge and the tide control the MXWL, is called a mixed-energy region'. The definitions of the River-dominated and Mixed regions are not clear and contradictory to what we see in Figure 7. If we consider 2 cases with tidal forcing off and on and discharge equaled to 3000 m^3/s, we clearly see, that the presence of tides influences the MXWL till 70 km from River mouth (Figure 7). So, based on this definition, the mixed region starts nearly from the border, where river source is prescribed. From another point of view, you can say, that the MXWL is defined by the river until MXVL reaches ~2m. Based on this definition you will get an extended River-dominated area compared to present in the manuscript, and this area will be getting larger increasing the discharge. There can be more definitions, please, clarify yours.

**Response:** *The suggestion has been followed. Firstly, we change the term "mixed-energy region" to the "transition zone". Then, according to the simulation result obtained with an open river boundary located further upstream, we moved the limit between the transition zone and discharge-dominated region to about 150 km from the river mouth. This point is consistent with the previous study* (Kästner et al., 2019)*, which found that at this point, the admittance of the tidal propagation upstream has a knickpoint, where dumping strongly increases. Here, we define the transition zone as the part of the river from where the MXWL profile between the "with-tides" and "without tides" scenarios start to differ to the area where discharge is no longer affecting MXWL profiles.*

**Technical:**
**Pg 3, 58-63:** Can be shortened to the point: The area of interest represents well-mixed and relatively shallow water body. Therefore, we applied 2D barotropic solution to reduce computational costs.
**Response:** *Suggestion has been followed*

**Pg 7, 172:** please, remove 'with satisfying frequencies' (you have already identified that in the beginning of the sentence, can be interpreted wrong).
**Response:** *Suggestion has been followed*

**Figure 8:** Please, leave only one subpanel, e.g., (a).
**Response:** *Suggestion has been followed*

**Table A3:** The details, which are the same for all experiments, can be identified in the Figure caption.
**Response:** *Suggestion has been followed*

**Figure 3:** The Figure is nice, but the font is really small compared to other pictures.
**Response:** *We updated the figure*

**Reference:**
Kästner, K., Hoitink, A. J. F., Torfs, P. J. J. F., Deleersnijder, E., and Ningsih, N. S.: Propagation of tides along a river with a sloping bed, J. Fluid Mech., 872, 39–73, https://doi.org/10.1017/JFM.2019.331, 2019.

**Response to the second comments of the second reviewer (RC2)**

We want to thank the reviewer again for taking the time to review our paper. His/her comments were really useful to help us further improve the manuscript.

**Minor comments**

**Nested model:** This approach is interesting. Is there a particular reason why the mesh of the large-scale SLIM model has not simply been refined at the drainage channels? Since SLIM allows for local mesh refinement and uses implicit time-stepping, I would not expect a large penalty on the runtime. Even with two models, why was HEC-RAS chosen over SLIM for the nested model?

**Response:** *Increasing the mesh resolution with the highest resolution of 10 m (much lower than 50 m) to cover the canals within the city, while its lowest resolution is 10 km, can nonetheless substantially increase the number of mesh elements and hence the computational time. HEC-RAS has further been specifically designed to simulate inundations based on a digital elevation map. In that respect, it is more mature than SLIM for such applications. Regarding HEC-RAS implementation, there is no particular reason to choose this model except it is because this model is well-known for tackling flood simulation over floodplain* (Loveland et al., 2021; Santiago-Collazo et al., 2019; Pasquier et al., 2019; Bush et al., 2022)*, and we have already used it previously.*

**Upstream boundary:** The authors explain that they placed the upstream boundary near Terentang about 100 km upstream from the sea, to avoid missing discharge from tributaries (mostly Tayan and Meliau) downstream of the head of tides at about km 300, near Sanggau. While this approach indeed reproduces the river discharge at the inflow boundary, it cuts the tidal prism, and thereby reflects the tidal wave and reduces the tidal discharge. The figure below shows the tidal discharge estimated with the theory of tides (Hill and Souza, 2006; Kastner et al., 2019). Truncating the domain as in the numerical model reduces the tidal discharge in the Kapuas Besar branch by 50% and increase the tidal discharge in the Kapuas Kecil branch by 30%. To get both the river and tidal discharge right the boundary could be moved to Sanggau while the inflow is set to the discharge measured at Terentang.

**Response:** *The reviewer made a very good suggestion and we followed it. We moved the upstream boundary to Sanggau. As a result, the MXWL profile changed, where the maximum water level at the Kapuas Kecil river mouth is equal to the tidal range. Then, according to the new result, we moved the limit between the transition zone (a new name for mix-energy region) and river-dominated region to about 150 km from the river mouth. At this point, the MXWL profile between the "with-tides" and "without tides" scenarios start to differ. This point matches with the previous study* (Kästner et al., 2019)*, which found that at this point, the admittance of the tidal propagation along the Kapuas River has a knickpoint, where dumping strongly increases. According to this new result, figure 7 and 9, and the related analyses in the text have been updated.*

**Bathymetry:** My comment on erroneously shallow cross-sections in the original manuscript was not clear. What I mean is not the mouth bar but that the raw bathymetry data of (Kastner et al., 2017) erroneously contains shallow cross-sections between Pontianak and the upstream

bifurcation. This is due to glitches of the echo sounder used for the measurement. The SLIM model results show jumps in surface elevation at km 30 and 45. This seems physically implausible and is probably due to backwater caused by erroneous constrictions of the cross-section. I suggest verifying this and if applicable, filtering the bathymetry along-channel.

**Response:** *We preprocessed the bathymetry and then compared it to the Garmin's and Pushidrosal's products. The bathymetry profile along the river is now consistent. Since the raw bathymetry made the simulation unstable, we smoothed the bathymetry map in our pre-processing step. This procedure guarantees that the model will run smoothly. So, we did the filtering process before using it in the simulation.*

**Terminology:** I agree with the first reviewer who commented that the adopted zone-terminology is somewhat confusing. The terms tidal energy and maximum water level are used interchangeably throughout the manuscript. However, there is no direct correspondence between the maximum water level and the (kinetic) energy. The maximum water level is a combination of the tidal amplitude and tidally averaged water level. The effect of tides on the mean water level is largest upstream of the point where most of the tidal energy has already been dissipated, as it integrates along channel, while the tidal amplitude decreases gradually along channel. The storm surge, furthermore, contributes an important part to the energy budget. Therefore, I recommend referring to water levels throughout the manuscript, and avoiding the term "energy".

**Response:** *The suggestion has been followed. We now avoid the term "energy" and redefine the mix-energy zone as the transition zone. We define the transition zone as the part of the river from where the MXWL profile between the "with-tides" and "without tides" scenarios start to differ to the area where discharge is no longer affecting MXWL profiles.*

**Typography**

**19** could divide -> divide: *The suggestion has been followed*

**105** For the wind shear stress a surface roughness is required, similar to cd for the bed shear stress. What value was chosen?

**Response:** *For wind speed less than 20 m/s, the wind drag coefficient (Cw) is defined by:*

$$Cw = 0.001*(a + b*U_{10})$$

*where, according to Smith and Banke (1975), a = 0.63 and b = 0.066. $U_{10}$ is the 10 m wind speed. For higher wind speeds, we set 0.003 as the saturated wind drag coefficient* (Moon et al., 2007).

*We added this information in the last of this paragraph: Here, the wind stress was computed with the Smith and Banke (1975) formula for the wind speed of less than 20 m/s and was computed with the Moon et al. (2007) formula for wind speed higher than 20 m/s.*

**188,189** new mesh -> second mesh: *The suggestion has been followed*

**194** 0.09m -> 0.09 m: *The suggestion has been followed*

**197** semidiurnal components explain the rest -> there is no rest (90.69 + 9.31 = 100): *we removed this sentence.*

**231** will drop -> drops: *The suggestion has been followed*

**234** leads to a reduction in the water levels -> reduced the water level: *The suggestion has been followed*

**236** not too significant -> not significant: *The suggestion has been followed*

**242** State in here that the reference for the 2.8 m water level is the lowest astronomical tide (LAT) and that the 2.8 m correspond roughly to 1.8 m above mean sea level and 0.7 m above highest astronomical tide (HAT). State also the river discharge for this day.
**Response:** *The suggestion has been followed. We added these sentences in the paragraph: The 2.8 m water level reference is the lowest astronomical tide (LAT). It corresponds roughly to 1.8 m above mean sea level and 0.7 m above the highest astronomical tide (HAT).*

**245** Please state the Kapuas discharge and tidal range (without storm surge) for that day!
**Response:** The suggestion has been followed. We added this sentence in the paragraph: *During the event, the Kapuas and Landak rivers had discharges of 4400 m3/s and 502 m3/s, respectively. At the river mouth, the tidal range reached 1.8 m.*

**251** top -> high water level? **Response:** *yes, it is. We changed word "top" to "peak"*

**263** Landak river streams -> Landak River: *The suggestion has been followed*

**277** the wind velocity less than 9 m/s or more than 24 m/s, it does not -> wind velocities less than 9 m/s or more than 24 m/s do not: *The suggestion has been followed*

**281** zone border -> boundary: *The suggestion has been followed*

**281** mix-energy -> mixed-energy: We change the term: "mix-energy" to "transition"

**282** border -> boundary: *The suggestion has been followed*

**293** from the river mouth to the upstream -> upstream from the river mouth: *The suggestion has been followed*

**293** was coincidentally met with a high river discharge -> I would call this more an intermediate discharge, as it seems to be less than 1/2 of annual peak discharge of the river.
**Response**: *The suggestion has been followed. Since the discharge of the Kapuas River is only 4400 m3/s, we agree that it is only an intermediate discharge. We changed the word: "a high river discharge" to "an intermediate river discharge".*

**321** where ebbs no longer impact -> where tides no longer impact: *The suggestion has been followed*

**Figure 7** It would be insightful to complement this figure with an along channel plot of tidal range and tidally averaged water level.
**Response**: *The suggestion has been followed. The figure has been updated.*

**Figure 13** Limit the colourmap of water depth between 0 m and 2 m, to better distinguish flooding in the city. **Response**: *The figure has been updated.*

**Reference:**

Bush, S. T., Dresback, K. M., Szpilka, C. M., and Kolar, R. L.: Use of 1D Unsteady HEC-RAS in a Coupled System for Compound Flood Modeling: North Carolina Case Study, J. Mar. Sci. Eng. 2022, Vol. 10, Page 306, 10, 306, https://doi.org/10.3390/JMSE10030306, 2022.

Kästner, K., Hoitink, A. J. F., Torfs, P. J. J. F., Deleersnijder, E., and Ningsih, N. S.: Propagation of tides along a river with a sloping bed, J. Fluid Mech., 872, 39–73, https://doi.org/10.1017/JFM.2019.331, 2019.

Loveland, M., Kiaghadi, A., Dawson, C. N., Rifai, H. S., Misra, S., Mosser, H., and Parola, A.: Developing a Modeling Framework to Simulate Compound Flooding: When Storm Surge Interacts With Riverine Flow, Front. Clim., 2, 35, https://doi.org/10.3389/FCLIM.2020.609610/BIBTEX, 2021.

Moon, I. J., Ginis, I., Hara, T., and Thomas, B.: A Physics-Based Parameterization of Air–Sea Momentum Flux at High Wind Speeds and Its Impact on Hurricane Intensity Predictions, Mon. Weather Rev., 135, 2869–2878, https://doi.org/10.1175/MWR3432.1, 2007.

Pasquier, U., He, Y., Hooton, S., Goulden, M., and Hiscock, K. M.: An integrated 1D–2D hydraulic modelling approach to assess the sensitivity of a coastal region to compound flooding hazard under climate change, Nat. Hazards, 98, 915–937, https://doi.org/10.1007/S11069-018-3462-1/TABLES/6, 2019.

Santiago-Collazo, F. L., Bilskie, M. V., and Hagen, S. C.: A comprehensive review of compound inundation models in low-gradient coastal watersheds, Environ. Model. Softw., 119, 166–181, https://doi.org/10.1016/J.ENVSOFT.2019.06.002, 2019.

Smith, S. D. and Banke, E. G.: Variation of the sea surface drag coefficient with wind speed, Q. J. R. Meteorol. Soc., 101, 665–673, https://doi.org/10.1002/QJ.49710142920, 1975.